# Interactions of Bacteriophages with Animal and Human Organisms—Safety Issues in the Light of Phage Therapy

**DOI:** 10.3390/ijms22168937

**Published:** 2021-08-19

**Authors:** Magdalena Podlacha, Łukasz Grabowski, Katarzyna Kosznik-Kawśnicka, Karolina Zdrojewska, Małgorzata Stasiłojć, Grzegorz Węgrzyn, Alicja Węgrzyn

**Affiliations:** 1Department of Molecular Biology, University of Gdansk, Wita Stwosza 59, 80-308 Gdansk, Poland; magdalena.podlacha@ug.edu.pl (M.P.); karolina.zdrojewska@onet.pl (K.Z.); malgorzata.stasilojc@biotech.ug.edu.pl (M.S.); grzegorz.wegrzyn@biol.ug.edu.pl (G.W.); 2Laboratory of Phage Therapy, Institute of Biochemistry and Biophysics, Polish Academy of Sciences, Kładki 24, 80-822 Gdansk, Poland; lukas.grabowski95@gmail.com (Ł.G.); k.kwasnicka@hotmail.com (K.K.-K.)

**Keywords:** bacteriophages, interactions between phages and eukaryotic organisms, phage therapy, prophages

## Abstract

Bacteriophages are viruses infecting bacterial cells. Since there is a lack of specific receptors for bacteriophages on eukaryotic cells, these viruses were for a long time considered to be neutral to animals and humans. However, studies of recent years provided clear evidence that bacteriophages can interact with eukaryotic cells, significantly influencing the functions of tissues, organs, and systems of mammals, including humans. In this review article, we summarize and discuss recent discoveries in the field of interactions of phages with animal and human organisms. Possibilities of penetration of bacteriophages into eukaryotic cells, tissues, and organs are discussed, and evidence of the effects of phages on functions of the immune system, respiratory system, central nervous system, gastrointestinal system, urinary tract, and reproductive system are presented and discussed. Modulations of cancer cells by bacteriophages are indicated. Direct and indirect effects of virulent and temperate phages are discussed. We conclude that interactions of bacteriophages with animal and human organisms are robust, and they must be taken under consideration when using these viruses in medicine, especially in phage therapy, and in biotechnological applications.

## 1. Introduction

The emergence of bacteria resistant to most or even all known antibiotics has become a serious medical and therapeutic problem. The impossibility of curing patients suffering from infections caused by multi-drug-resistant strains of bacteria has led to the establishment of the term “antibiotic crisis” [1]. The problem is global because antibiotic-resistant bacteria, including pathogenic strains, are found worldwide [2], contributing to enormous difficulties in medicine [3]. Therefore, it is necessary to find alternative methods of treatment of human and animal diseases caused by bacteria [4].

Bacteriophage therapy (phage therapy) is a potential method of combating bacterial infections by using bacteriophages—viruses that can replicate in bacterial cells and eliminate them [5]. There are several advantages of using phages to treat infected humans and animals. These include the specificity of phages toward selected bacteria (without affecting the natural microbiota), the restriction of phage propagation to the presence of their specific hosts, and the ability to kill antibiotic-resistant bacteria [6]. Furthermore, the current literature lacks strongly documented significant adverse effects associated with phage administration, and phage therapy is widely regarded as safe. However, there are also controversies regarding both the efficacy and safety of this therapeutic method [5,6]. Both older and recent papers indicated that interactions between bacteriophages and eukaryotic cells are possible, affecting both biochemical and physiological processes. Therefore, one may wonder whether such interactions are actually safe, considering that despite a lack of acute reactions, some milder effects might be still deleterious for patients or animals subjected to phage therapy procedures. If the human body is treated as an ecological habitat, bacteriophages can occur not only in the gastrointestinal tract, which is obvious due to gut bacteria present there, but also in the blood, urine, or cerebrospinal fluid, which could already theoretically pose potential risks [7]. For example, it was suggested that circulating phages may effectively interact with the host immune system [8]. However, this impact is still being investigated and discussed. Studies on the mouse model using the T4 bacteriophage did not confirm its immunomodulatory properties [9].

An interesting fact is the interaction between the different types of bacteriophages and human cellular matrix molecules, such as fibronectin, gelatin, and heparin [10]. These types of dependences are crucial, as they can directly translate into changes in tissues and organs function. Intriguingly, phages were demonstrated to be able to enter mammary epithelial cells through endocytosis and even reach nuclei. Such a mechanism of phage penetration may facilitate the transportation of bacteriophages through epithelial cell layers. Indeed, the transcytosis process has been proposed to be responsible for this phenomenon [11]. Another report demonstrated that phages can internalize neuroblastoma cells through endocytosis and perhaps might also enter the nucleus [12]. It appears that terminal proteins (enzymes that can prime the DNA replication of some bacteriophages) have fragments resembling nuclear localization signals and thus may penetrate to the nuclei of eukaryotic cells [13]. In this light, the modulation of expression of some eukaryotic genes by bacteriophages might be considered likely. Our knowledge on possible interactions of bacteriophages with eukaryotic cells, mammalian tissues, distribution of bacteriophages throughout human body, and interaction of phages with mammalian immune and nervous systems has been summarized and discussed in this review article.

## 2. Ways for Penetration of Animal and Human Tissues by Bacteriophages

### 2.1. The Epithelial Barrier

Bacteriophages can penetrate layers of epithelial cells and possibly spread to different areas of the body, including blood, lymph, internal organs, and even the brain. Penetration strategies and mechanisms vary and depend primarily on the type of bacteriophage. It was reported that the proposed mechanisms include the following: “Trojan horse”—by which the phage-infected bacterium enters the epithelial cells or is absorbed by them; “phage display”—which requires homing ligands to be placed on viral capsids to regulate receptor-mediated cell recognition and endocytosis; and “free uptake” of phage particles by eukaryotic cells via endocytosis [11]. For all of these mechanisms, there is supporting and contradictory evidence, suggesting that phages can reach the body via different routes. The T4 phage binds weakly to the mucin glycoprotein using the immunoglobulin-like domain of the Hoc protein. The weak binding maximizes the phage’s ability to kill bacteria by allowing it to move across mucosal surfaces in a sub-diffusive manner. Such a sub-diffusive movement allows the T4 phage to explore specific regions of the mucus [14]. In turn, Bille et al. [15] reported an interesting interaction of a commensal bacteria *Neisseria meningitidis* with epithelial cells, which is essential for colonization of the human *nasopharynx*, and in some cases can cross the blood–brain barrier. They showed that the presence of prophage of filamentous phage, designated MDAϕ for Meningococcal Disease Associated, through viral production, increases meningococcal colonization on epithelial cell monolayers. The detailed research revealed that meningococci are bound to the apical surface of host cells by several layers of highly piliated bacteria, while in the upper layers, the bacteria are not piliated but surrounded by phage particles. The latter case probably corresponds to bacteriophages during their extrusion through the outer membrane. As the phenomenon increases, the loss of piliation in the upper layers does not allow aggregation by forming bundles of phage filaments attached to bacterial cell walls.

Another interesting mechanism is transcytosis, which is defined as the transport of macromolecular cargo from one side of a cell to the other within membrane-bounded carriers [16]. Nguyen et al. [11] used in vitro assays to demonstrate the rapid and directional transcytosis of various bacteriophages through cell layers originating from gut, lung, liver, kidney, and brain. The transcytosis of bacteriophages through cell layers showed significant preferential directionality for transport from the apical to basal side with approximately 0.1% of all bacteriophages within a 2 h period. Microscopic and cytological analyses revealed that bacteriophages reached both the vesicular and cytosolic compartments of eukaryotic cells, with phage transcytosis suggesting transport through Golgi apparatus mediated by the endomembrane system. These results allowed estimating that 31 billion bacteriophage particles are transcytosed through intestinal epithelial cell layers into the human body every day. At the same time, this mechanism explains the ubiquitous presence of bacteriophages in eukaryotic organisms.

Cellular internalization can occur through a number of pathways that vary depending on the ligand and type of internalized cell. Tian et al. [17] described the internalization mechanism of M13 phage. The filamentous M13 phage was found to be highly dependent on the cell type in its interaction with cells and the mechanism of internalization. This phage tends to bind on the cell membrane of only epithelial cells and not endothelial cells. In addition, M13 phage enters cells by endocytosis with a specific mechanism: clathrin-mediated endocytosis and macropinocytosis for MCF-7 (differentiated mammary epithelium cells) and caveolin-mediated endocytosis for human dermal microvascular endothelial cells (HDMEC). In turn, Kim et al. [18] described the internalization mechanism of the M13 phage, which was mediated by cell-penetrating peptides, such as 3D8-VL or TAT. In the first case, the process takes place through caveolae-mediated endocytosis by interacting with heparan sulfate and proteoglycans as cell surface receptors, while TAT-decorated M13 phage has been internalized by caveole-mediated endocytosis using chondroitin sulfate, suggesting that phage internalization occurs by a physiological endocytotic mechanism via specific cell surface receptors rather than non-specific transcytotic pathways. After endocytosis, the internalized phage particles were localized in the endosomal compartments, the endoplasmic reticulum, and the Golgi apparatus within 6 h. This may again indicate a Golgi-mediated transcytotic pathway.

However, a growing body of evidence suggests that phages are in fact capable of binding specific receptors on the cell surface, allowing internalization by endocytic vesicles without target ligands. Lehti et al. [12] described the binding and penetration of *Escherichia coli* PK1A2 bacteriophage into live eukaryotic neuroblastoma cells in vitro. The phage interacts with a polysialic acid on the cell surface that has structural similarity to the bacterial phage receptor. Based on microscopic analysis, internalization was shown to occur via the endolysosomal pathway and resulted in phages persisting inside the cell for up to 1 day without adversely affecting cell viability. The authors highlighted the possibility of other epitopes on the eukaryotic cell surface, which show structural similarity to polysaccharides present on bacterial hosts, to be receptors for phages. Many studies emphasized the specific role of β3 integrins in this type of interaction. A possible molecular mechanism for these effects has been proposed, involving a specific interaction between the Lys-Gly-Asp motif of the phage protein 24 and β3-integrin receptors on target cells. Anti-β3 antibodies and synthetic peptides mimicking β3 natural ligands have also been shown to inhibit phage binding to cancer cells. This is consistent with the well-described integrin β3-dependent tumor metastasis mechanism [19]. Moreover, one indirect evidence of internalization is the presence of homologs of fragments of various genes in phages and eukaryotic cells. Substantial evidence for DNA sequences associated with genes found in bacteriophages of the *Microviridae* family not only in various prokaryotic organisms, but also in eukaryotic cells has been reported [20]. Conversely, the presence of bacteriophages in obligate intracellular bacterial parasites of eukaryotes may promote DNA bidirectional transfer [21]. This could have potentially dangerous consequences, especially from the point of view of the wider use of phage therapy.

### 2.2. The Circulatory System

Regardless of the route of administration used, the presence of bacteriophages in the blood is confirmed relatively quickly. This is primarily due to their ability to move across the endothelial cell barriers. In the study by Bochkareva et al. [22], rectally administered phages against *Pseudomonas aeruginosa*, *Salmonella enteritidis*, and *Escherichia coli* were detected in blood samples at all investigation time points (30, 45, 60, 75 min and 3, 6, 9 h). Two detection methods, microbiological agar-layer technique and PCR, confirmed the presence of phage DNA in blood samples collected from the rabbits, with the probability increasing between 3 and 6 h after suppository administration, depending on the type of phage. However, the factors that did not affect the presence of phage particles in the collected blood samples were morphology and taxonometric parameters. Capparelli et al. [23] confirmed the stable persistence of the phage фD lytic for *Escherichia coli* O157:H7 in the mouse circulatory system for at least 38 days. The described phage was isolated from bovine manure and had characteristics of both *Myoviridae* (contractile tail) and *Syphoviridae* (presence of the *msp* gene). In addition to its high stability in the circulatory system, it showed the ability to eliminate bacteria in mice within 48 h of intragastric administration. In turn, Yasuhiko and Toshihiro [24] reported the ability of some phages, particularly the PPpW-4 phage against *Pseudomonas plecoglossicida* in goldfish, to penetrate the intestinal wall into the circulating blood within just 10 min after oral administration. The persistence time of these phages in the circulatory system was up to 12 h, indicating a promising therapeutic potential in combating bacterial infections after oral administration. The phenomenon involving the rapid movement of phages into the circulating blood can have a number of functional consequences. Due to the increasing number of studies on this issue, the term “phagemia” has already started to be used in the literature. The presence of bacteriophages in serum was already confirmed in the 1970s [25,26]. Chu et al. [25] tested 37 bovine sera samples for the presence of phages. They were positive in 23 cases. The number of plaque-forming units (PFU) per ml of serum varies from 1 PFU per ml to 10^4^ PFU per ml. Orr et al. [27] confirmed the presence of Gram-positive bacteria (*Bacillus* sp. and streptococci) and bacteriophages in bovine serum when used in vitro as a cell culture medium. Although the biological implications of the presence of bacteriophages in bovine serum for in vitro studies are not clear, the long-term persistence of bacteriophages that had infiltrated bovine serum into supplemented cell cultures was confirmed. It is worth highlighting the fact that phages were isolated from bovine fetal serum samples, which had no bacterial hosts. This suggests that the “phagemia” phenomenon may be common in the serum of healthy organisms [28].

The question arises as to whether the presence of phages is widespread in the bloodstream and whether they can exert specific effects on the functioning of the entire body. Analyzing the available literature data, it emerges that the results are still inconclusive. Mankiewicz and Béland [29] demonstrated the presence of mycobacteriophages in sera from 75% of patients with sarcoidosis, whereas they were unable to isolate these phages from the sera of healthy individuals or patients with tuberculosis. However, other researchers not only confirmed the presence of the above-mentioned phages in serum collected from 19 Crohn’s disease patients but also in four healthy volunteers, aged 18 years [30]. Undoubtedly, the route of administration is an important determinant of phage penetration and persistence in peripheral blood. In this regard, rectal administration is the most effective route of administration that results in the presence of the greatest number of phage particles in the bloodstream in all animals. When administered intramuscularly, phage titers were high in the short term, but the most rapid increase was observed within several minutes after injection [31]. An important point is the ability of coliphages to adhere to erythrocytes as well as leukocytes [32]. The titer of these phages increases significantly in rabbit plasma on the 4th day after intragastric administration. The opposite observation was made by Keller and Engley [33], who did not confirm the adhesive capacity of T1 coliphage circulating in peripheral blood. An important element that cannot be overlooked in such considerations is the role of specific receptors and ligands. It was investigated what particular signal peptides might mediate the mechanism of bacteriophage translocation [34]. Using immunocytochemical analysis, the transport of M13 phage bearing the YPRLLTP peptide across the intestinal barrier, along specific channels, was shown. Experimental confirmation of specific protein-dependent phage transport across different types of the body’s natural barriers might have beneficial effects on increasing their bioavailability regardless of the administration route and the effectiveness of phage therapy.

### 2.3. The Endothelial Barrier

Phages that have entered the circulatory system by infiltrating endothelial cells reach the organ in which they ultimately exhibit therapeutic activity. The fact that bacteriophages can directly interact with the barrier structure formed by endothelial cells was confirmed by in vitro and in vivo studies. Møller-Olsen et al. [35] conducted research on the effectiveness of phage therapy in the context of combating infections with antibiotic-resistant bacteria that cause neonatal meningitis. The applied model was based on *E. coli* EV36, bacteriophage K1F, and human cerebral microvascular endothelial cells (hCMECs). It was observed that the described bacteriophage is phagocytosed in a PAMP-LC3-dependent manner, which does not result in elevated inflammatory markers (TNF-α, IL-6, IL-8, IFN-β), yet it affects the permeability of the endothelial barrier, which may facilitate the penetration of immune cells into the endothelial vessel. The high degree of heterogeneity in the structure and molecular features of the vascular endothelium was described as early as the 1990s. Rajotte et al. [36] tested the molecular diversity of phage homing peptides by targeting the vasculature of different organs and tissues. They demonstrated their specificity, even versatility, making them capable of acting as molecular addresses to facilitate the interaction of different types of bacteriophages. In turn, based on the phage display method, it was indicated that hierarchical forms of peptides, rather than monomeric ones as previously thought, determine their interactions with various cell types, including cerebral endothelial cells, which directly translates into the degree of permeability across the blood-brain barrier and thus interaction with neurons or glial cells [37]. This type of research, especially if confirmed in animal models, is very important in terms of identifying molecules that can act as carriers of various pharmaceuticals that will exhibit therapeutic effects directly in the brain. Considering the structure of the endothelial barrier, size and shape are very important determinants of permeability. As it is well known, the endothelium separates blood and tissues, which are formed by cells with an ordered but heterogenous structure. Whether the particular molecules can be transported across this barrier and have a direct effect on the target organ also depends on the physiological state of the body. This happens differently in a healthy organism and when inflammation develops. In the first case, only molecules up to 70 kDa can be transported, whereas as a result of disease and elevated concentrations of inflammatory markers, when the structure loosens, molecules as large as 2000 kDa can penetrate [38]. This is supported by observations of patients who have elevated levels of phage DNA in their blood in the course of diseases associated with immune dysfunction [39]. This is likely an effect of pro-inflammatory cytokines that enhance transport across the endothelial barrier, which is particularly evident in bacterial diseases [40]. In terms of shape, spindle-shaped or cylindrical particles are more easily permeable than spherical ones. In contrast, under fluid flow conditions, spheres and short micelles are taken up by cells more readily than longer filaments [41]. Berkowitz and Day [42] described the filamentous phage fd that, despite its size of 14,600 kDa, had a high rate of movement (average axial distance 3.82 ± 0.15 Å) across the endothelial barrier in various species. In the transport of various molecules and bacteriophages to target organs or tissues, the role of surface elements of the extracellular matrix cannot be overlooked. Fibronectin, gelatin, or heparin can capture and bind various proteins of the phage capsid, facilitating or impeding its penetration. Another important factor that cannot be overlooked in such considerations is the wide variability in endothelial structure. Depending on the type of organs surrounded, the pore size can vary from 62–68 nm up to 200 nm in diameter, which allows penetrations into the liver or bone marrow [30]. Despite the fact that phages belong to many families, differing in morphology, genetic material, or molecular characteristics, most of them could move freely across the endothelial barrier, especially in organs with a high degree of vascularization and blood supply, such as the brain, which offers great opportunities from the point of view of developing new therapies.

### 2.4. The Blood–Brain Barrier

A particular type of barrier is the blood–brain barrier. Its permeability is strictly regulated, which on the one hand has a beneficial effect on the protection of the crucial organ, the brain, but on the other hand makes it difficult to penetrate therapeutics that could reduce the negative effects of neurodegeneration. Over 98% of potential drugs for diseases resulting from nervous system dysfunction are rejected due to lack of permeability across the blood–brain barrier. Thanks to the development of nanotechnology, molecules are being created whose structures are based on similarity to the bacteriophage capsid and whose transport mechanisms are based on the Trojan horse strategy mentioned earlier. Anand et al. [43] described a functional nanoparticle that, through genetic and chemical manipulation, enabled the modification of the phage P22 capsid to become a carrier for analgesic drugs. Biocompatibility and permeability by endocytosis was confirmed in vitro by the use of the PBMVEC-BBB model and human microvascular endothelial cells as well as in vivo in mice. An example of a novel approach in the treatment of Alzheimer’s disease was the use of filamentous phage that, through the use of genetic engineering methods, exhibited a single antibody chain on its surface that enabled it to penetrate the blood-brain barrier and through its immunomodulatory properties reduced the formation of β amyloid plagues [44]. However, this property of phage was first described in 1943. Dubos et al. [45] demonstrated the presence of bacteriophages against *Shigella dysenteriae* in mouse brain as early as 1 h after intraperitoneal administration. Moreover, the high titers persisted for a long time. The surprising ability of the filamentous phage M13 was reported, which despite its size (900 nm) easily penetrated the blood-brain barrier within a short time after intranasal administration [44]. After application of 10^11^ phage particles, the highest titer of M13 phage was detected in the hippocampus and olfactory bulb of mice. According to the authors, the high ability of this phage to penetrate the brain is mainly due its linear shape and structure. Subsequent studies by Ksendzovsky et al. [46] showed that the mechanism of active axonal transport allowed the M13 phage to move freely through the gray and white matter. Thanks to modern methods, it was possible to create a data bank (phage display method) targeting specific bacteria, as well as to modify the protein components of their capsids to obtain higher concentrations and favorable pharmacokinetic parameters in target organs, including the brain, which directly translates into increased efficiency of the developed phage therapies [39]. Among the literature data from both cellular and animal studies, numerous confirmations of the long-term activity of the homing peptides can be found. The modification involving increased expression of transferrin receptors resulted in more efficient infiltration in an orthotopic mouse model of glioblastoma multiforme [47]. Urich et al. [48] described novel transport vectors identified by performing in vivo phage selection using a rat model, based on cannula implantation into the cisterna magna—a reservoir containing cerebrospinal fluid that was collected to evaluate the efficacy of the selected vectors. The biological activity of the peptides used, which were introduced into the capsid of phage T7, was verified by using the BACE1 inhibitor, which through binding to newly developed protein transporters led to a 40% reduction in β amyloid level in cerebrospinal fluid. This type of data clearly demonstrated that with proper selection of transport vectors, it is possible to achieve therapeutic effects even in organs that are difficult to access or are immunologically privileged such as the brain.

### 2.5. The Skin

When considering various aspects of bacteriophage interactions with eukaryotic organisms, the majority of publications describe experiments in which phages were applied orally or intravascularly. In contrast, relatively little is known about the effects of skin administration and methods of further propagation. The human skin is a multi-layered protective barrier. Bacterial infections of the skin and soft tissues vary in etiology and severity. Statistically, 7 to 10% of patients hospitalized for other diseases develop skin infections, which are often accompanied by an increased immune system response manifested by high fever as well as elevated local and systemic inflammatory markers. Due to the large diversity of organisms colonizing the skin, diagnostic difficulties, as well as possible complications with oral therapy, the potential for the use of phages in this type of infections is enormous, but it requires a greater understanding of the mechanisms of interaction and penetration through the skin layers [49]. While one can find quite a few documented descriptions of the successful treatment of bacterial skin infections using phages, the mechanism of penetration of deeper skin layers is not well documented. Kumari et al. [50] compared the efficiency of silver nitrate with gentamicin and Kpn5 phage applied topically in a hydrogel to a wound infected with *Klebsiella pneumoniae.* The results obtained clearly showed that even a single administration of hydrogel with Kpn5 phage protects mice against the development of bacterial infection, while such an effective prophylactic effect is not observed even with repeated applications of silver nitrate with gentamicin. The high efficacy when administered to the wound surface is due in part to the fact that the bactericidal action of the phage was not limited by the host immune system. Consequently, the bacteriophage was not only released from the hydrogel but also able to penetrate the wound, eradicate the target bacteria, and prevent the development of septic shock, which can be a direct cause of death.

The composition of the microbiome has a large impact on the effectiveness and skin penetration ability of bacteriophage. This relationship has been described for a relatively long time. Keller and Engley [33] showed great variability in the data for the presence of bacteriophages against *Bacillus megatherium* on mouse skin that had not been previously shaved or exposed to mechanical trauma. This was probably due to the activity of the natural antibodies, properdin system, as well as the fact that the morphology and size of this and other bacteriophages described (T4 or those against *Staphylococcus*) are similar to many animal viruses. Pitol et al. [51] compared the degree of skin permeability of bacteriophage MS2 and two enteric viruses. The transfer between the liquid and different types of skin (synthetic, collected from living volunteers and deceased) was analyzed. The concentration of virus in the liquid from which propagation occurred and the thickness of the layers as well as the structure of the skin surface layer have been shown to be important factors. Understanding the relationship between microorganisms inhabiting the human skin and their interactions with the host organism as well as discovering the possibility of pharmacological manipulations of these interactions is crucial from the point of view of seeking therapy to combat and prevent dermatological diseases [52].

The summary of barriers that must be crossed by phages to penetrate tissues and organs of animals and humans is summarized in Figure 1.

## 3. Interactions of Bacteriophages with Mammalian Immune System

As indicated in the preceding chapter, bacteriophages may interact with the surfaces of mucosa, penetrate the epithelial cell layer, and spread throughout the body [53]. Moreover, bacteriophages may interact with cells of the immune system, leading to a cytokine response and induction of phagocytosis. Due to the nucleoproteinaceous structure, bacteriophages are recognized by cells of the immune system, leading to their neutralization and clearance from the animal or human organism. Additionally, they may modulate the adaptive immune response, which results in the production of anti-phage antibodies [8]. The hypothetical model of bacteriophage interactions with various components of the immune system was proposed by Van Belleghem et al. [54].

### 3.1. Antiphage Innate Immune Response

Innate immune response is the body’s first line of defense against microorganisms. The components of the innate immune response include phagocytes (dendritic cells and macrophages), granulocytes (basophils, eosinophils, neutrophils, mast cells, and natural killer cells), and complement system proteins that increase phagocytic uptake by phage opsonization [8]. Furthermore, cells of the innate immune system are capable of recognizing microorganisms, including bacteriophages. The interactions of various components of the innate immune system with bacteriophages are described below.

#### 3.1.1. Dendritic Cells

Dendritic cells (DCs) are antigen-presenting cells and originate from the bone marrow myeloid progenitor cells [55]. Immature forms of these cells are capable of phagocytosis [56], as first demonstrated by Barfoot et al. [57]. However, they are also capable of activating adaptive immune response by presenting antigens to T lymphocytes in lymph nodes. DCs are present in tissues that come in contact with the external environment [55].

An et al. [58] examined the effects of bacteriophage ES2 on the expression of surface proteins CD86, CD40, and MHCII, the production of pro-inflammatory cytokines IL-6, IL-1α, IL-1β, and TNF-α by dendritic cells, and the activation of the NF-κB signaling pathway. The authors showed that this bacteriophage increased the expression of surface proteins as well as pro-inflammatory cytokines. Furthermore, they observed that there was the activation and translocation of NF-κBp65 to the nucleus, leading to the activation of NF-κB signaling [58].

Miernikiewicz et al. [9] investigated the influence of bacteriophage T4 on the ability of dendritic cells to synthesize pro-inflammatory interleukins as well as changes in the expression profile of these cell surface proteins. No significant effect of bacteriophage T4 on the production of cytokines IL-1α, IL-6, IL-12, and TNF-α, as well as on the expression of MHC class II, CD40, CD86, and CD80 was observed [9]. Similar results were obtained by Freyerberger et al. [59] who tested whether bacteriophage K affects cytokine expression and activation of human dendritic cell markers. They showed that bacteriophage K had little or no effect on the production of anti- and pro-inflammatory cytokines and the expression of MHC-I and CD80/CD86 proteins [59].

Bocian et al. [60] examined the effects of bacteriophages T4 and A3/R on human myeloid dendritic cell differentiation. The authors showed that bacteriophages did not affect the process of dendritic cell differentiation, as well as their role in T-lymphocyte activation [60].

#### 3.1.2. Monocytes and Macrophages

Monocytes are produced in the bone marrow and then enter the bloodstream, where they circulate for 2 to 3 days before entering tissues and transforming into macrophages. The monocyte population accounts for 3–8% of all leukocytes in the peripheral blood. Active macrophages are found in the liver (Kupffer cells), spleen, connective tissue (histiocytes), central nervous system (microglia cells), and lungs (alveolar macrophages). The main function of macrophages is to degrade microorganisms, synthesize cytokines, and present antigens to lymphocytes, leading to activation of the adaptive immune response [55]. Phagocytosis is the main process leading to the removal of bacteriophages from the organism either directly by macrophages or indirectly (e.g., by opsonization) [55]. The entire process is complex and involves membrane remodeling, receptor motion, cytoskeletal reorganization, and intracellular signaling [56].

The first study on the ability of macrophages to phagocytose bacteriophages was presented as early as in 1964 by Aronov et al. [61]. The authors observed that bacteriophage T2 was phagocytosed by macrophages [61]. The degradation of bacteriophages by macrophages was also confirmed for phages λ, P22, and φX174 [62].

Yıldızlı et al. [63] tested the effects of two bacteriophages infecting *E. coli* on the activation status of mammalian macrophages and TNF-α levels. The bacteriophages were able to effectively activate macrophages to produce TNF-α in the absence of lipopolysaccharide (LPS), which is a bacterial stimulant of the inflammatory response. However, the number of phage particles per ml was similar in experiments with phages administered alone and in combination with LPS. This indicated that under these conditions, there were no effects of LPS on macrophage phagocytic potential. The most probable explanation for this phenomenon was the activation of macrophages by bacteriophages through a signaling pathway different than LPS [63].

#### 3.1.3. Granulocytes

Granulocytes, also called polymorphonuclear leukocytes, are the group of leukocytes produced in the bone marrow that have characteristic granules in their cytoplasm and a segmented cell nucleus [55]. Mature neutrophils represent about 40–70% of all leukocytes in peripheral blood. They are the first inflammatory cells to migrate toward the site of infection by bacteria. Neutrophils are phagocytes capable of combating microorganisms through a combination of action of reactive oxygen species that are excreted from cells, thus weakening bacteria and facilitating their internalization, and hydrolytic enzymes that are secreted into the phagosome [64].

A recent study Roach et al. [65] examined the effect of bacteriophage PAK_P1 on the response of human neutrophils. They showed that the application of bacteriophage at a high concentration of 10^9^/mL induced IL-8 production. However, no induction of apoptosis of resting neutrophils was observed, nor did it induce expression of the surface CD11b protein or lead to an oxidative burst [65]. These results were similar to those obtained by Borysowski et al. [66,67], who showed that the exposure of bacteriophage A3/r to neutrophils did not lead to an oxygen burst and did not cause neutrophil degranulation.

Eosinophils are another group of granulocytes that represent about 2–3% of all white blood cells. These cells persist in the circulation for about 8–12 h, while in the absence of stimulation, they are able to survive in tissues for 8–12 days. Although the key role of these cells is to control parasitic infections, an increasing number of studies indicates that they are involved also in defense against microbes [68]. Basophils are responsible for inflammatory reactions during acute and chronic allergies. These cells produce histamine, serotonin, and Il-4 [69]. However, data on the impact of bacteriophages on eosinophils and basophils are scarce. Chen et al. [70] observed no significant increase in eosinophilic basophils in *Pasteurella multocida*-infected mice that received phage therapy.

#### 3.1.4. Foreign Particles vs. Innate Immune Response

Cells of the innate immune system, such as dendritic cells and macrophages, express receptors called pattern recognition receptors (PPR) that are able to recognize different pathogen-associated molecular patterns (PAMPs). This family of receptors include Toll-like receptors (TLRs) and nucleic acid receptors such as cyclic-di-nucleotide (CDN) sensors [71]. Upon recognition of the target ligand, such as viruses, foreign nucleic acids, LPS, and flagellin, PPRs induce pro-inflammatory or anti-inflammatory response [8,72]

TLRs are the best characterized class of PPRs [73]. TLR1, 2, 4, 5, 6, and 11 recognize pathogens extracellularly, while TLR3, 7, 8, and 9 recognize pathogens endosomally [72]. Viral nucleic acids are recognizable by TLR3 (double-stranded RNA), TLR7 and TLR8 (single-stranded RNA), and also TLR9 (DNA) (Figure 2). Moreover, TLR9 recognizes unmethylated CpG in prokaryotic genomes, leading to the activation of the innate immune system and release of TNF-α, IFN-γ, IL-2, IL-6, and IL-8 [74,75].

In a recent study, Gogokhia et al. [76] showed that bacteriophages activated the synthesis of IFN-γ through TLR9. In the studied germ-free mice groups, which were treated with purified bacteriophages (3 × 10^7^ PFU/mL), a significant increase in the level of IFN-γ was observed, as well as a significant increase in the level of CD8+ T cells. Moreover, it has been observed that the stimulation of dendritic cells by bacteriophage DNA can induce IFN-gamma production by CD4+ T cells [76]. This suggests that bacteriophages may affect several cells of the immune system.

Lee et al. [77] investigated the effect of bacteriophages on the levels of IFN-γ, pro-inflammatory cytokines, and heat shock proteins in hens infected with *Salmonella* Typhimurium. It has been shown that in *Salmonella*-infected chickens treated with bacteriophages, mRNA expression of IFN-γ, TLR4, and IL-4 in the jejunum, as well as IFN-γ, HSP27, and TNF-α in the liver were decreased relative to the *Salmonella*-infected but non-treated control [77]. On the other hand, Zeng et al. [78] tested the effects of bacteriophage supplementation on TLR2, TLR4, and TLR9 levels in piglets and found that the supplementation of 400 mg/kg bacteriophages increased the mRNA expression of tested TLRs in the jejunum. These results proved that bacteriophages activated the immune system by regulating the inflammatory response through TLRs [78].

#### 3.1.5. Clearance of Bacteriophages

Bacteriophages are able to activate the immune system, which leads to their rapid removal from the body. This is one of the main problems in maintaining the appropriate phage titer in therapy [79]. It is known that bacteriophages can be phagocytosed by cells of the immune system. The major organs involved in filtering circulating bacteriophages are the liver and the spleen. They contain a system of mononuclear phagocytes (MPS), which is the group of specialized cells that are responsible for neutralizing bacteriophages [79]. It is assumed that the clearance time of bacteriophages in animal and human organisms may depend on many variables, such as the size of phage virions, the dose of bacteriophages used, or the method of their administration [80].

It is believed that the half-life of bacteriophages in the body may depend on the dose used or the amount of phage particles accumulated in organs and cells, but the data are not entirely consistent. Various half-lives of bacteriophages in mammalian organisms were reported, from 2.3 h, through 4.5 h, to 8 h [81,82,83].

It was suggested that larger phage virions are easier to filter out than smaller virions. However, due to the insufficient number of studies, it is not possible to fully determine the influence of bacteriophage morphology on the clearance time [80,84].

Studies on the effect of bacteriophage encapsulation on the pharmacokinetics of bacteriophages were published. Namely, the use of encapsulation protected bacteriophages and allowed their prolonged circulation in the body, making them less visible to the immune system and reducing their susceptibility to neutralization [80].

The liver and spleen are considered to be the main organs involved in bacteriophage filtration and clearance. In these organs, bacteriophage titers are usually the highest [85,86,87,88,89]. Moreover, the retention time of active phages in liver and spleen is often the longest [19]. The liver and spleen contain a group of specialized mononuclear phagocytes called Kupffer cells in the liver. Although both these organs efficiently filter phage virions, it is believed that the phagocytes in the liver inactivate bacteriophages more rapidly [80].

Kaźmierczak et al. [90] examined the circulation of fluorescent bacteriophages in a mouse model. After intravenous administration, the highest number of bacteriophages was observed in the liver. However, they observed that bacteriophages were present in the spleen 60 min after injection, while they were not detectable in the liver [90]. This may suggest that Kupffer cells play a key role in the rapid and efficient removal of bacteriophages from the body.

The clearance time of bacteriophages in the body also depends on the properties of their surface proteins. Thus, an attempt has been made to create bacteriophages with modified surface proteins, which could theoretically attenuate their immunogenicity. However, effective control of the circulation of bacteriophages with modified proteins has not been confirmed [91]. Nevertheless, there are reports confirming that the pharmacokinetics of such bacteriophages has been altered, but this information is not entirely conclusive. Merill et al. [92] tested the ability of MPS to neutralize bacteriophage λ with a mutation in the capsid E protein. They found that the mutant phage persisted in the body 24 h longer than the wild-type counterpart. The replacement of glutamic acid by lysine in the E protein, resulting in a charge change, might be a reason for this phenomenon. As a consequence, the modified bacteriophages were less susceptible to recognition by MPS [92]. Another explanation for this process could be the decreased susceptibility of bacteriophages to the complement system, as the exposure of arginine or lysine on the surface of the phage capsid facilitated bacteriophage escape from the host immune response. Similar results were obtained by Vitiello et al. [93], who showed that exchanging glutamic acid with lysine in the E protein increased the survival of bacteriophage λ 1000-fold. However, a recent study by Hodyra-Stefaniak et al. [94] showed that altering proteins on the bacteriophage capsid can have the opposite effect. They engineered mutants of bacteriophage T4 that exposed seven types of peptides on the capsid. No accumulation of modified bacteriophages T4-B and T4-G2 was observed in organs, while the titers of the three modified mutants decreased 100-fold relative to the wild-type phage. In addition, it was observed that modified phages were significantly more strongly inactivated by the complement system than wild type, whereas no changes in phage sensitivity to phagocytosis or immunogenicity occurred [94]. Lysine was present in the peptides of both mutants, while arginine was absent in either mutant. Thus, the authors suggested that the reason for a stronger stimulation of the complement system by the modified phages was a change in the amino acid composition of the capsid [94].

#### 3.1.6. Phage-Induced Cytokine Response

To confirm the ability of bacteriophages to directly induce a cytokine response, an endotoxin-free preparation is required. Several studies indicated that bacteriophages directly modulate cytokine responses, both pro-inflammatory and anti-inflammatory [79].

Chen et al. [95] investigated the immunomodulatory abilities of bacteriophage vB_SauM_JS25 in bovine mastoid epithelial MAC-T cells. The study showed that this bacteriophage reduced levels of TNF-α, IL-1β, IL-6, IL-8, and IL-10. Moreover, the suppression of LPS-induced phosphorylation of the nuclear factor NF-κB was observed [95]. This indicated an anti-inflammatory effect of bacteriophage vB_SauM_JS25. Similar results were obtained by Miernikiewicz et al. [96], who examined the effect of the gp12 protein on the immunomodulatory capacities of bacteriophages in a mouse model. They showed that the protein caused almost complete depletion of IL-1α and reduction of the IL-6 level by 50% [96]. Moreover, Xue et al. [97] observed that the levels of pro-inflammatory IL-6, TNF-α, and Il-1β significantly decreased in mice treated with bacteriophage X1.

Pjanova et al. [98] investigated the effect of bacteriophage-derived dsRNA on cytokine response in blood mononuclear cells. They showed that bacteriophage-derived dsRNA induced the synthesis of pro-inflammatory IFN-γ, IL-1β, and IL-6 [98]. Khan Mirzaei et al. [99] tested the ability of *E. coli* bacteriophages to stimulate cytokine responses in blood mononuclear cells and epithelial cells. These authors noted a significant increase in levels of pro-inflammatory cytokines IL-8, CXCL-1/GROα, and macrophage migration inhibitory factor (MIF) in HT-29 cells while observing the release of IL-6, IL-10, and TNF-α in blood mononuclear cells. Van Belleghem et al. [54] also performed an expression analysis of immune-related genes in peripheral blood monocytes induced by staphylococcal bacteriophage and four *Pseudomonas* phages. They showed that *Pseudomonas* bacteriophages induced the expression of pro-inflammatory genes (coding for CXC1, CXC5, IL-1α, and IL-1β) but also anti-inflammatory genes (encoding IL-1RN and IL-10) [54].

However, different results were obtained in studies on effects of purified bacteriophage T4 and capsid proteins gp23*, gp24*, Hoc, and Soc on stimulation of inflammatory responses in a mouse model and in cells isolated from human blood. Levels of IL-1α, IL1-β, IL-2, IL-6, IL-10, IL-12 p40/70, IFN-γ, TNF-α, MCP-1, MIG, RANTES, GCSF, GM-CSF, and reactive oxygen species (ROS) were examined, but no significant effects of T4 bacteriophage and its capsid proteins on inflammatory cytokine production and ROS levels could be observed [9].

In summary, the effect of bacteriophages on the induction of cytokine responses may depend on many factors: first, the level of purification of the preparation from LPS, which is an extremely potent stimulator of the immune response. However, other factors such as type of bacteriophage, the route of administration, the length of treatment, and the production of anti-phage antibodies should be taken into consideration [100].

#### 3.1.7. Phages as Factors Increasing Bacterial Phagocytosis

The use of bacteriophages in therapy may increase the phagocytosis of bacteria by macrophages. It is believed that bacteriophages opsonize the bacterium, making them more recognizable by immune cells, leading to effective elimination of the bacteria. Additionally, during phagocytosis, bacteriophages continue their lytic development, acting synergistically with the immune system to combat the bacterial infection [101].

One possible mechanism of phagocyte response to bacteria is the production of ROS. However, excessive ROS production can induce oxidative stress and tissue damage. Break et al. [102] investigated the effect of phage T4 on ROS levels. The authors showed that this bacteriophage affected the phagocytic system and caused a small increase in ROS. However, in response to *E. coli* infection, the phage inhibited ROS production. It seems that this phenomenon was not caused by the direct action of the bacteriophage but by a reduction in the number of bacteria due to lysis by the phage [102]. On the contrary, in vitro studies suggested that bacteriophages can also induce an increase in ROS production [103]. Effects of the lytic bacteriophage EFA1 levels on secreted ROS and the growth of *E. faecalis* in a co-culture with HCT116 colon cancer cells were evaluated. There was a significant decrease in bacterial cell number but also an increase in ROS levels in HCT116 cells co-cultured with *E. faecelis* [103]. It seems that differences in ROS induction are dependent on cell type.

### 3.2. Antiphage Adaptive Immune Response

The main task of the adaptive immune response is to specifically recognize and eliminate the pathogen but also to produce memory cells to prevent reinfection or damages. The main cells involved in this process are lymphocytes, which have the ability to recognize different cell structures, distinguish small differences between them, and exhibit immune memory. However, for activation of adaptive immunity, the action of innate immunity is necessary [104].

Bacteriophages as foreign protein-nucleic particles will also be recognized by the immune system, leading to the production of anti-phage antibodies. Bacteriophages are capable of inducing the production of different classes of antibodies (Figure 3). The most common initial step is the production of IgM in response to the first administration of a phage. After repeated phage administration, an increase in IgG level is observed. There is also an increase in IgA level, which builds the immune barrier on the surface of mucous membranes [105]. However, depending on the protein composition of the capsid and tail, bacteriophages vary in immunogenicity and can induce different antibody responses, which depends on the route of administration. Moreover, naturally occurring bacteriophages can induce humoral responses [105]. The testing of antibody levels against phage T4 revealed the presence of IgG specific for phage proteins Hoc, Soc, gp23*, and gp24* in 81% of sera [8,106]. Analysis of antibodies against *Pseudomonas* phages in healthy human population was also performed, and the presence of antibodies neutralizing bacteriophages LMA2 (in 11% of cases), F8 (in 15% of cases), and DP1 (in 40% of cases) was detected in sera [107].

Many factors influence the effectiveness of antiphage antibody production, including the route of administration and duration of therapy [108]. Majewska et al. [109] examined the impact of two staphylococcal phages on induction of antibodies production in the gut and blood. Bacteriophages were administered orally to mice for 100 days; then, bacteriophage application was stopped for 120 days, after which bacteriophages were again administered to mice for 44 days. It was observed that both bacteriophages induced the production of IgA, IgG, and IgM antibodies in the blood and IgA in the gut. Moreover, IgM levels were highest after 22 days, while IgG levels increased before the end of the experiment. It was also observed that IgA levels decreased after the end of phage apposition. Oral administration of bacteriophages induced a weak response to phages [109]. It was also examined whether bacteriophage T4 can induce humoral response in the gut and blood of mice to observe that bacteriophage-induced antibody production when the time of administration was sufficiently long. An increase in levels of IgG antibodies occurred after 36 days of application, while levels of IgA antibodies increased after 79 days. Moreover, compared to the first application, the re-administration of bacteriophages led to a faster secretion of IgA [110].

Recent studies demonstrating antiphage antibody production by mammals are summarized in Table 1.

## 4. Interactions of Bacteriophages with the Respiratory System

### 4.1. Phage Therapy against Bacterial Infections of Lungs

Regardless of the route of administration, because of its rich vascularization, the lung is an organ with relatively high accession for bacteriophages. Phages can reach this organ either by oral, intranasal, or intravenous administration (see also Figure 1). It was shown that phage therapy is most effective against respiratory infections after intranasal application or tracheal delivery [115]. As early as in the 1960s, the efficacy of an aerosol formulation that contained the T-2r phage was described [116]. Debarbieux et al. [117] described the effectiveness of bacteriophage PAK-P1 against *Pseudomonas aeruginosa* in controlling lethal infection in mice but also its prophylactic potential when administration occurred 24 h before the lungs become infected with bacteria. The rapid course of the bacterial eradication reaction by bacteriophage in the lung indicates the absence of a cellular intermediate factor in this mechanism, such as protease activity. Interestingly, the effectiveness of bacteriophages is also determined by the course of the infection, especially whether it is acute or chronic. In addition, it was observed that despite previous reports indicating that bacteriophages are rapidly eliminated from the human body, this was not confirmed for the lungs. The administration of phages even 72 h before bacterial infection has been shown to be effective in completely preventing symptoms.

Despite promising results, the stability of phage formulations must be taken into account. Dry inhalable powders were proposed as a solution [118]. Dufour et al. [119] observed several highly clinically relevant advantages of using phage therapy to control acute pneumonia in mice that was induced by the intranasal administration of two strains of *E. coli* (536 and LM33). Two selected bacteriophages (536_P1 and LM33_P1) and the most commonly used antibiotics (ceftriaxone, cefoxitin, or imipenem—Cilastatin) were selected for comparison. The efficacy of both groups of therapeutics was tested at two time points, reflecting the progression of infection. Additionally, the phages were also administered to healthy animals to control the level of immune system response. Despite the rapid course of bacterial cell lysis after treatment with both bacteriophages, no excessive stimulation of the innate immune response was observed. In addition, phage therapy allowed the normalization of peripheral blood parameters, and the application of phages to healthy animals resulted in a slight increase in the levels of released cytokines (IFN-γ, IL-12) and chemokines, which have antiviral potential. These effects were noticed only in the lungs but not in the blood. In contrast, Roach et al. [118] emphasized the specific role of synergy between neutrophils and bacteriophages, which is crucial in combating bacterial lung infections. In the above study, infection was induced by administering *P. aeruginosa* to groups of mice with different immune deficits. Regardless of the infection strategy, neutrophils play a key role, and the bacteriophages used are well tolerated in the lung and are not neutralized by immune effector cells. The main role in the described synergy is played by B and T lymphocytes rather than by non-specific lymphoid cells residing in the mucosa, especially when chronic infections are considered. Further studies are needed to answer the question of how such interactions can proceed when immune cell status changes and their percentage increases. This may be a factor that inhibits the bactericidal activity of phages, especially when the time between the infection and the applications of phage therapy is too long.

### 4.2. Penetration of Respiratory System by Phages in the Light of Anti-COVID-19 Therapy

Phage therapy is not only a promising method to combat antibiotic-resistant bacteria: perhaps it can be a missing link in the fight against the COVID-19 pandemic. Some compounds or drugs effective against other diseases have shown high potential against different coronaviruses in vitro, but there are no confirmed reports on their efficacy and safety in vivo. Considering all these aspects and the difficulty in selecting effective antibiotics, phage therapy seems to be a method that can fill the gap.

Numerous reports support the antiviral potential of phages, phage lysates, or their genetic material, mainly via cellular receptors or the enhanced release of different groups of interferons [120]. Patients infected with SARS-CoV and SARS-CoV-2 often exhibit apoptosis and lymphocytopenia, the massive nature of which often leads to death [121]. One of the facts supporting the use of phages in virus control is their ability to penetrate epithelial cells, to protect against apoptosis, as well as to regulate the expression genes coding for chaperone proteins (heat shock proteins), which directly affect the virus life cycle and offset the negative effects of its propagation. As described earlier, the KGD motif, present in the protein capsid of T4 phage, is an important element in the interaction between eukaryotic cells and bacteriophages. It is presumed that this sequence determines the blockade of adsorption of various viruses (including coronaviruses) to lung epithelial cells. Another important aspect is the correlation between the severity of symptoms of infection induced by respiratory viruses and levels of oxidative stress markers, which are significantly increased in SARS-CoV-infected cells. In this regard, phages also seem to be useful, because by inhibiting the transcription of genes coding for NF-κB, they also reduce toxic ROS production, especially in the affected organism [120]. Wu et al. [121] published optimistic reports on the use of phage therapy in four critical COVID-19 patients with an additional bacterial infection. This creates new opportunities for the treatment of patients who are struggling with severe infections that are not only bacterial but also viral. They are presented schematically in Figure 4.

## 5. Interactions of Bacteriophages with Mammalian Central Nervous System

### 5.1. Use of Phages in the Treatment of Brain Diseases

Various features of bacteriophages have been employed to use these viruses as potential drugs for neurological diseases. The feasibility to obtain genetically modified M13 phage displaying a truncated single-chain form of an antibody against a β-amyloid (Aβ) fragment and to deliver it to the central nervous system (CNS) has been used. This allowed for not only in vivo detection but also the disaggregation of β-amyloid plaques in a transgenic mouse model of Alzheimer’s disease [44,122]. Nevertheless, phage M13 itself has an interesting property that makes it a potential agent for use in neurodegenerative diseases such as PD and AD. In a cellular model, an inhibitory effect was observed in the formation of α-synuclein (AS) aggregates, which is a hallmark of Parkinson’s disease [123,124]. The mechanism of this phenomenon is not yet fully understood, but it is speculated to involve preferential binding of the phage to the N-terminal fragments of AS fibers.

In addition to the modulatory effect of filamentous phages, a reduced amount of total AS was observed, which is indicative of AS clearance after treatment with filamentous phages [123,124].

Highly purified preparations of native M13 were demonstrated to be able to bind to and cause the disruption of a variety of misfolded protein assemblies, including Aβ, α-synuclein, tau, and yeast prion Sup35 [125]. The characterization of amyloid fiber binding and remodeling indicated that the bacteriophage minor capsid protein, gene product 3 protein (gp3), is critical for this activity. The two N-terminal domains of gp3 that facilitate the binding and disruption of amyloids have been defined as a general amyloid interaction motif (GAIM) [125].

Filamentous phage fd has been used to treat cocaine addiction [126]. The phages were genetically modified to present single-chain antibodies against cocaine on the surface of the capsid. Intranasally administered antibody-conjugated phages blocked the psychoactive effect of the drug in a mouse model [126].

Members of the Ff filamentous bacteriophages family (among them fd, f1, and M13) were reported to possess anti-tumorigenic properties. Wild-type M13 phage can stimulate cultured tumor-associated macrophages (TAMs) to polarize their activity to anti-tumorigenic M1 phenotype and promote the migration of cytotoxic neutrophils in response to factors secreted by stimulated TAMs [127]. The in vivo treatment of mice bearing subcutaneous melanoma tumors with tumor-specific phages (displaying peptides targeted to mouse melanoma) led to an intense anti-tumorigenic response associated with neutrophil infiltration into the tumor microenvironment and prolonged survival [128]. Hence, the intranasal delivery of Ff phages was applied in an aggressive murine model of glioblastoma. Ff phages accumulated in the brains of mice and inhibited brain tumor progression [129]. However, it could not be excluded that the anti-tumorigenic property of the applied phages might be due to the presence of trace amounts of LPS in phage preparation. Therefore, Ff phages might be “carriers” of LPS, and its amount could be sufficient to trigger anti-tumorigenic immune response.

### 5.2. Phages as Central Nervous System Pathogens

Over the past decades, evidence has emerged that the gut microbiome can influence behavior and nervous system health. Intestinal bacteria possess the capacity to produce numerous neuroactive molecules, such as serotonin, catecholamines, glutamate, γ-amminobutyric acid (GABA), and short-chain fatty acids (SCFAs). Disruption of the gut microbiota homeostasis, so-called dysbiosis, may increase intestinal permeability and bacterial translocation, determining an immune system’s overresponse and consequent systemic and/or CNS inflammation [130].

In this context, bacteriophages may play an indirect role in diseases of CNS through influencing gut microbiota. In 2018, Tetz et al. [131] raised the hypothesis that the presence of large amounts of lytic *Lactococcus* phages in the microbiota of patients with Parkinson’s disease may be associated with this neurodegenerative disease. This may be related to the decreased level of *Lactococcus* spp. in the patients’ group. These bacteria are considered a source of microbiota-derived neurochemicals, including dopamine, which they produce in appreciable physiological amounts. However, it is hard to determine if the decrease in the production of intestinal dopamine may be associated with early gastrointestinal symptoms of Parkinson disease or involved in triggering the neurodegenerative cascade of the disease [131].

Another example of an indirect association of phages with CNS diseases was provided by Yolken et al. [132]. In that study, a single phage, *Lactobacillus* phage ϕadh, was identified that was significantly more prevalent and abundant in patients with schizophrenia than in healthy controls. The primary bacterial host for phage ϕadh is *Lactobacillus gasseri*, which is a common component of the oral and gastrointestinal mucosa and is known to play a role in the modulation of intestinal permeability [132].

The presence of bacteriophages in the cerebrospinal fluid (CSF) has been observed and possibly associated with the pathogenesis of multiple sclerosis (MS). Tetz et al. [133] claimed to detect the presence of *Shigella* phage SfIV and *Staphylococcus* phage StB2 in the CSF of patients with MS, compared with 15 control patients with other neurological diseases [133]. However, patients with MS are characterized by increased intestinal permeability and a disrupted blood–brain barrier, suggesting that the phage presence in the CSF may not be a cause but rather a result of the disease [134].

## 6. Bacteriophages in the Gastrointestinal Tract

Bacteriophages are a part of the animal gastrointestinal microbiome [135]. They are thought to be responsible for shaping microbiome composition and bacterial diversity as well as facilitating horizontal gene transfer [135,136]. However, although phages mainly interact with bacteria, they also interact with eukaryotic cells of gastrointestinal tract (GIT) [136,137]. These interactions may be direct or indirect, when compounds released by lysed bacteria or phage proteins cause a reaction in the system [76,138]. Aside from naturally occurring phages, those introduced during phage therapy may occasionally occur in GIT. Oral phage therapy was proven to be effective in farming animals and humans [139,140]. Therefore, the influence those phages have on microbiome, as well as intestinal cells, has been a subject of intensive studies in order to determine if the use of oral phage therapy can be regarded as safe for patients [139,141].

While describing phages of animal and human GIT, it is usually the phages present in the small and large intestine that are considered. Phages can be found in the stomach, usually if they have been previously orally introduced to the system. However, since most phages are sensitive to pH lower than 2, only a small percentage survives [139,142]. It was observed that some phages penetrate to the blood from the stomach; however, the process has low effectivity, and phages mainly pass through to small intestine [139].

Bacteriophages colonize the intestines of an infant alongside bacteria, and the virome composition stabilizes within first years of life [143]. The number of phages in human GIT is estimated at ≈10^10^/g, and the virus-to-microbe ratio (VMR) is thought to be ≈1:10, which is lower than in other known ecosystems [136]. It was believed that most phages present in human GIT are temperate phages existing in gut bacteria in the form of prophages [136,144]. Most of them are representatives of *Caudovirales* order and *Myoviridae* family [136]. CrAss-like phages are also described as one of the most abundant phage groups in GIT. These *Podoviridae* family representatives infect bacteria belonging to *Bacteroidetes* phylum and show high stability and prevalence in the gastrointestinal phage population over time [145,146]. However, recent analysis indicated that the statement that temperate phages dominate the gut in healthy individuals may not be accurate. It has been suggested that there is no ‘core’ set of phages in the human gut, as predominant clusters can drastically vary from one individual to another [136,147,148]. The type of phage, its form, and life cycle influence interactions with both bacterial and animal/human hosts. Free phage particles may directly interact with eukaryotic cells in GIT, triggering various responses [76,136], or they can disrupt natural gastrointestinal flora [149,150]. On the other hand, prophages may influence eukaryotic organism indirectly i.e., by encoding toxin genes produced by bacteria and horizontal gene transfer [151,152,153].

### 6.1. Phage Adherence to Mucus Layer

Gastrointestinal mucosa is a natural habitat of various microorganisms that form a complex community and interact with their eukaryotic host [154]. Most microbial residents grow within the mucus layer, acting as the first line of defense against pathogens [155]. Bacteria may utilize pili, fimbriae, excrete specific proteins, and promote biofilm formation in order to facilitate adhering to the epithelial layer of the intestine [155,156]. In vitro experiments showed that bacteriophages are also able to adhere to mucosal surfaces and that by doing so, they may modulate microbial colonization and pathogenicity [138]. An experiment involving T4 bacteriophage and various mucus-producing tissue cell lines showed that the bacteriophage was able to bind glycan residues displayed on mucin glycoproteins via capsid proteins and more precisely by Ig-like domains present in Hoc capsid protein [138]. Those types of highly antigenic Ig-like domains were also observed in other members of *Caudovirales* order [157,158]. It is now proposed that aside from biding to bacterial surface carbohydrates during infection, the Ig-like protein scaffold may adapt to the host’s changing patterns of mucin glycosylation and increase prevalence in the mucus layers [138,158]. Mucus-adhered bacteriophages in the intestine have a higher probability of encountering the bacterial host than free virion particles. Then, it was speculated that mucus adherence may help shape the gastrointestinal microbiome and prevent pathogens from colonizing the system [14,136,138].

### 6.2. Phage Translocation from Gut to Bloodstream and Other Organs

One of the key questions regarding oral phage therapy is when phages can enter the bloodstream and other organs and whether it has any health implications [139]. Phages are known not to infect eukaryotic cells due to a lack of phage-specific receptors on the cell surface and incompatibility between prokaryotic and eukaryotic replication, transcription, and translation systems. However, phages have been shown to be able penetrate bodies of higher vertebrates with ease, using various mechanisms (see Section 2) [139,159,160]. Therefore, it is not uncommon for orally administered phages to enter systemic circulation from the intestine and spread to other organs [139,159]. In vitro studies demonstrated that phage transcytosis across confluent cell layers has a preferential directionality for apical-to-basal transport. The proposed general mechanism suggests that phages access the endomembrane compartments of an eukaryotic cell. Then, phages are enclosed in vesicles and pass through the Golgi apparatus before being exocytosed [11]. It has been shown that only 0.1% of phages survives such transport, with some phages remaining inside of the cell [11]. Therefore, it is assumed that this is one of the ways phages can enter the bloodstream from gastrointestinal system. The other possible way was suggested to be more direct, with phages crossing through punctured vasculature and damaged epithelial cell layers at sites of inflammation caused by disease, bacterial toxins, or phage and bacterial DNA [133,136]. When phages cross the epithelial layer of the intestine and enter the bloodstream, they can spread throughout the body [139]. It has been observed that phages can reach other organs of the digestive system, such as the liver and spleen, and some were even detected in cerebrospinal fluid [133,136]. However, there are also reports that no phages have been found in blood samples of animals treated with oral phage therapy [139]. Therefore, it is assumed that penetration to the bloodstream as well as to other organs may be phage-specific as well as depend on the individual treated [139,159].

### 6.3. Inflammatory Bowel Disease and Other Diseases Influenced by Phages

The gut microbiome has been revealed to have an influence not only on the health of GIT. It was linked with stimulating the immune system, immunity development, and can even affect brain biochemistry [161,162]. The pathogenesis of various human diseases is nowadays associated with alterations in gastrointestinal microbiota [163]. One of the best described examples of phage interactions with gut eukaryotic cells in disease is inflammatory bowel disease (IBD) [76,133,164]. The research on IBD and its complications has shown a number of ways that bacteriophages can influence the course of illness.

Phages can have a direct influence on the course of IBD, as shown by Gogokhia et al. [76]. In their study, they observed that bacteriophages and phage DNA stimulated IFN-γ via the TLR9 receptor. This led to a heightened immune response and inflammation in mice. Furthermore, an increase in bacteriophage levels exacerbated colitis via the same pathway [76]. The study on ulcerative colitis patients also revealed that phages isolated from their fecal samples induced more IFN-γ than phages isolated from healthy individuals. Fecal microbiota transplant was shown to help decrease the inflammation and relieve the illness symptoms [133,165]. However, phages can influence mammalian health in more than just one way. Apart from directly acting as inflammatory agents, phages can affect the health of their host by changing the intestinal microbiome. In patients with Crohn’s disease and type 1 and 2 diabetes, changes in the phagobiome resulted in changes in the microbiome, and that those changes could result in the severity of disease symptoms [166,167,168]. Those changes in the microbiome could result in increased intestinal barrier permeability or ‘leaky gut’, which would result in phages, bacterial debris, proteins, and DNA passing to the bloodstream and further increasing inflammation [133,136]. It was also hypothesized that this prolonged, severe inflammation in the gastrointestinal tract could also lead to an increase in the induction of prophages. This proposal is corroborated by observations that the lytic to lysogenic phage ratio is higher in patients with IBD, and more temperate phages in virion form were isolated from samples of patient feces [135,136,164]. This shows that relationships between phages and their environment inside GIT could be very complex. Moreover, the interactions between phages and the intestinal epithelium and cytokines could have a direct influence on the mammalian host, but interactions between phages and bacteria could also affect the course of the disease. All those factors can cumulate, and therefore, the role of phages in IBD and other intestinal diseases should be studied taking those multiple interactions into account [76,133,135,136,141,166].

### 6.4. Bacteriophages as Gastrointestinal Tract Pathogens

While studying the role of phages in IBD and other diseases, it was observed that phages can influence the course of the disease both directly and indirectly. It was also observed that the oral administration of phages could alter microbiota even in healthy individuals. This has led to a concept of phages as mammalian pathogens that can be a source of infection [133,169,170]. Phage DNA and RNA act as inflammatory agents that trigger immune response [171]. Furthermore, phage-induced bacterial lysis can lead to increase in levels of cell-free DNA as well as bacterial pathogen-associated molecular patterns (PAMPs) such as LPS, peptidoglycan, and bacterial amyloid in gut and bloodstream [133,172,173]. The resulting inflammation may lead to an increase in gastrointestinal wall permeability and the spread of inflammatory agents in an organism, leading to an illness [133,135]. The “phage infection” may occur with the oral consumption of lytic phages or bacteria carrying prophages with food or water or by the induction of prophages already present in GIT (for example, due to antibiotic therapy). This may lead to a shift in gut microbiota, elevated levels of PAMPs, and the recognition of inflammatory agents by TLR3, TLR7, TLR8, or TLR9, which induce the production of type I IFN that eventually leads to an autoimmune response or illness (Figure 5) [133]. The pathogenic influence of gastrointestinal phages has been observed in some neurogenerative diseases and immune-mediated disorders i.e., Parkinson’s disease or multiple sclerosis, as it is discussed in detail in Section 3 and Section 5.

### 6.5. Phage-Related Bacterial Toxins

Another way phages can influence the health of their eukaryotic host is by toxin release during prophage induction. Temperate phages are known to encode toxins that causes diseases of GIT, such as Shiga toxin or cholera toxin [174]. Under favorable conditions, phages lie dormant in the form of a prophage inside their bacterial host. However, following an induction event, the phage will initiate the transcription of its genome resulting in toxin production and release during the lysis of bacterial cells [135]. Antibiotics, such as fluoroquinolones, or hydrogen peroxide, produced by white blood cells, are known factors affecting the induction of prophages and toxin release [175].

One of the toxins encoded by phages is Shiga toxin, whose genes are present in the genomes of some of the lambdoid phages [151]. The toxin consists of two subunits, A and B, and it is one of the AB_5_ toxins [176]. The mechanism of action involves B subunits of the toxin binding with glycolipid globotriaosylceramide (Gb3) on the eukaryotic cell membrane. This binding causes an induction of narrow tubular membrane invaginations, which results in the formation of membrane tubules. Then, the toxin is transferred to the cytosol via the Golgi apparatus and endoplasmic reticulum. Once inside the cell, the A subunit is cleaved into two parts, with the A1 component able to bind to the 60S subunit of the ribosome, cutting off one A residue in 28S rRNA and halting protein synthesis. This leads to a hemorrhage, as the toxin mainly interacts with the lining of the blood vessels [152,177]. This results in abdominal pain and bloody diarrhea in patients [152].

Cholera toxin is another example of phage-related toxin. Virulent strains of *Vibrio cholerae* are known to carry a filamentous bacteriophage CTXφ [178]. This ssDNA phage is a member of the *Inoviridae* family, and it is shown to carry the gene of cholera toxin as well as two others [179]. The B subunit of the cholera toxin binds to GM1 gangliosides on the surface of eukaryotic cells. Once bound, the entire toxin is endocytosed by the cell, and the cholera toxin A1 chain is released by the reduction of a disulfide bridge. The endosome is moved to the Golgi apparatus, the ER, and into the cytoplasm by the Sec61 channel. Then, it is free to bind with ADP-ribosylation factor 6 (Arf6), which results in exposition of the toxin active site and enables its catalytic activity. The toxin catalyzes ADP-ribosylation of the alpha subunits of G proteins. This results in a chain reaction that causes an increased concentration of cAMP and over-activation of cytosolic PKA. Then, these active PKA phosphorylate chloride channel proteins, which leads to an efflux of Cl^−^ and to the secretion of H_2_O, Na^+^, K^+^, and HCO_3_^−^ into the intestinal lumen. Hence, rapid fluid loss from the intestine occurs, which is expressed clinically as a severe diarrhea that can lead to dehydration and death within a few hours from the first symptoms [180,181]. Apart from the cholera toxin gene, additional genes coding for two other toxins have been recently identified in the CTXφ genome: accessory cholera enterotoxin (Ace)—presumably a minor coat protein of virion stage CTXφ, and zonula occludens toxin (Zot). However, the role these toxins play in the virulence of cholera remains unclear [179].

Apart from bacteriophages carrying toxin genes, they can also indirectly modulate toxin production in some bacterial species. For example, studies on *Clostridium difficile*, an emerging nosocomial pathogen causing opportunistic intestinal infections, showed that prophages can indirectly modulate toxin production [175]. It has been shown that *C. difficile* lysogens carrying temperate phages ϕC2, ϕC6, ϕC8, and ϕCD38-2 displayed increased production of TcdA and TcdB toxins. The molecular mechanism by which phages affect toxin gene expression remains unknown [175,182]. However, this ability of phages to regulate the toxicity of its bacterial host may lead to the hypothesis that in years of coevolution, some phages seem to have integrated within the regulatory network of their host without losing their individuality [174,175].

### 6.6. Complexity of Phage-Mediated Effects in Gastrointestinal Tract

Studies on phages of GIT revealed a complex network of interactions between bacteria, bacteriophages, and eukaryotic cells of the system (Figure 6). Albeit the molecular details of some processes remain unclear, it is known that phages are an integral part of the gastrointestinal system environment [135,136,141]. They can have a beneficial impact on microbiota and their eukaryotic host [14,138], but they can also act as pathogens by increasing inflammatory response and toxin release, leading to illnesses [133,179]. However, in healthy gut, they seem to be a part of a stable ecosystem and to play an important role in the maintenance of gut homeostasis. On the other hand, some external conditions may result in phages having a negative impact on the well-being of the eukaryotic host [135,136,147]. The impact of externally introduced phages via oral administration or microbiota transplant on animal and human health is also the subject of intensive studies due to the spread of antibiotic resistance. It was reported that in most cases, phages had a positive impact on health or that no negative impact has been observed [139,141,159,160]. However, the interactions between phages and gastrointestinal microbiota and cells depend on a variety of factors such as the type of phage and individual subjected to treatment; thus, the full extent of influence that phages could possibly have on health may never be fully known [136,139,141].

## 7. Bacteriophages in Urinary Tract

The urinary tract of healthy mammals has long been considered a sterile environment. In the last few decades, new diagnostic methods have shown that this niche is inhabited by many microorganisms, including bacteriophages.

The presence of bacteriophages accompanying pathogenic bacteria was studied and described nearly a hundred years ago [183]. It was observed that bacteriophages were present in the urine samples of 25% of patients suffering from urinary tract infection (UTI) caused by *E. coli*. Moreover, *E. coli* cells susceptible to phage lysis were present also in 25% of samples but not necessary in the same samples. However, urine samples that came from healthy patients without any symptoms of infection never contained phages [183].

On the other hand, the only method of phage detection at that time was based on using susceptible bacterial strains and observation of clearing of a turbid culture and/or formation of plaques on solid media. Molecular methods or electron microscopy were not yet available.

Bacteriophages isolated from the examined urine sample were used in phage therapy of patients with UTIs. The results of this therapy were remarkable, as it was reported that phages were able to treat an acute infection in pregnant women or sepsis in 8-month-old infants [183]. At that time, it was assumed that the only way that bacteriophages might appear in the urinary tract was the induction of a prophage in lysogenic bacteria [184].

Another study of clinical urine samples from patents with UTIs revealed the presence of infectious phages in nearly half of the samples tested [185]. In some of these samples, phages were directly observed by electron microscopy. This suggests that the phages were present in the examined samples at a density of at least 10^7^ particles/mL, as this is the minimal density that allows the observation of phages by this method. That study revealed also a possible interference of phages with bacteria present in urine samples [185]. Although Gram staining of the urine sediments suggested an abundant presence of Gram-negative bacteria at a density greater than 10^5^ cells/mL, growth on the agar plate was less effective, suggesting a bacterial density of 10^3^ cfu/mL. One of the colonies obtained on agar plates was further used for the antibiogram agar plate testing, which revealed the presence of lysis plaques consistent with phages [185]. It is worth mentioning that the presence of phages in urine samples can prevent the detection of bacterial infection of the urinary tract, leading to misdiagnosis. Importantly, some of the samples containing phages revealed no bacterial presence [185]. Therefore, in contrast to opinion [183], it appears that phages can enter the urinary tract alone, independently from bacteria.

With the development of culture-independent techniques, such as new generation sequencing, it has become possible to study the microbiome and the virome in the body niches that previously were thought to be sterile. Several such studies have been performed in recent years revealing that the urinary tract is inhabited not only by bacteria but also by viruses. The majority of the viral sequences found in the samples belonged to phages, but some of eukaryotic viruses were also identified [186,187]. Another study revealed the presence of seven phages in four urine samples obtained from women with urge urinary incontinence. However, there is a lack of evidence of any relations of these phages with the conditions of patients [188].

Metagenome analyses of 30 urine samples were performed, in which samples were collected via catheterization from 10 healthy women and 20 women suffering from overactive bladder. Twelve of these 30 samples contained sequences predicted to be partial or complete viral genomes. Some of them exhibited sequence homology to previously characterized lytic or lysogenic phages infecting bacteria belonging to the genera *Streptococcus*, *Lactobacillus*, and *Gardnerella* [189,190].

Several studies showed the presence of prophages in the vast majority of bacterial isolates from the urinary tract. One may speculate that lysogeny is a survival strategy for phages in an environment where the bacterial density is low. The presence of a prophage in the genome can also be beneficial for bacteria, as it may protect lysogenic bacteria against superinfection. Prophages often encode virulence factors, and they may also contribute to the improvement of motility of bacterial calls [191]. Recent in vitro studies have shown that some prophages can be induced by changes in the pH of the environment [192]. It is worth mentioning that in patients suffering from urinary tract infections (UTIs) caused by *E. coli*, urine often has pH values below 6 [193]. These results may support the hypothesis that phages travel to the urinary tract as prophages in bacteria.

Recently, a newly identified filamentous phage UPφ901 has been isolated as a prophage from a clinical *E. coli* strain present in a patient’s urine sample [194]. As in many other filamentous phages, the genome of UPφ901 integrates into the dif locus of the host’s genome. Bioinformatic analysis revealed the presence of highly similar sequences in genomes of many strains belonging to various bacterial species, including *E. coli*, *S. enterica*, *K. pneumoniae*, *C. koseri*, and *Y. enterocolitica* [194]. It was speculated that filamentous phages can have a major impact on bacterial virulence by increasing cell motility and biofilm formation [194].

Phages were demonstrated to have anti-inflammatory effects, thereby reducing the production of ROS by neutrophils that can damage epithelia [195]. These findings were confirmed by studies of phage therapy in mice with UTI, where the presence of phages resulted in a decreased expression of pro-inflammatory cytokines, such as TNFα and MCP-1 [196].

Phage stability in urine can vary depending on different factors and conditions. The T3 phage was found to be stable in urine samples, even after a long time of incubation in urine and in hydrolyzed urine [197]. On the other hand, coliphage MS2 has a low survival ratio in urine as its titer dropped below the detection levels within one to three weeks of incubation in urine samples at 30 °C [198]. Other studies suggested that phages infecting urinary tract pathogens are adapted to low pH values and urine components [199]. Phages against *K. pneumoniae* were tested as a potential preparation for use in phage therapy. No decrease in the viability or infectivity of phages was demonstrated when they were incubated in urine samples [199]. Similar results were obtained in studies on the effectiveness of phages against *Enterobacter cloacae*. These phages showed stable viability and efficacy after at least 12 h in urine and were capable of reducing bacterial titers in urine by two orders of magnitude [200].

The kidneys and urinary tract play major roles in removing phages present in the blood. Radioactively labeled phages, when injected intravenously into mice, passed into the urine very quickly. After only 5 min, 17% of the injected phages were present in the urinary bladder. After 30 min and after 3 h, such a fraction was as high as 30% and >50%, respectively [201,202]. The level of phage excretion thorough the urinary tract is sufficient to achieve a therapeutic effect in the treatment of UTIs by the oral administration of phage preparations. It is claimed that phages are filtered in the kidneys not only through the Malpighian tufts but also through transfer from the blood to the canal epithelium [203].

UTIs are often caused by bacteria capable of colonizing and invading the epithelium present in the urinary tract. In vitro studies indicated that phages are capable of infecting bacterial cells adhered to the epithelium [204]. It was shown that bacteria adhering to the epithelium are up to 100 times less sensitive to antibiotics than bacterial cells in suspension, but when using bacteriophages, the difference between the susceptibility to therapy of bacteria adhering to the epithelium and those present in suspension was significantly smaller [204].

One of the concerns of phage therapy of UTIs may be the possible washout of phages during bladder emptying. However, according to the computational model, this should not be a problem due to phage self-replication, which is dependent on the bacterial load at the site of infection. Even when complete washout occurs in a relatively short time (0.5 h) after phage inoculation, it has a relatively small effect on the ability of the phage to reduce the bacterial population [205]. Moreover, evidence was presented that phages are capable of invading the mucous layer of the epithelium, and therefore, they are protected from the full washout [138].

One of the major causes of UTIs are catheters, which act as foci for biofilm formation. The most common biofilm-forming bacterium associated with urinary catheters is *P. aeruginosa*. Hence, there have been attempts to prevent biofilm formation by pretreating catheters with benign *E. coli* and phages against *P. aeruginosa*. In one study, the synergistic effect of benign *E. coli* HU2117 and *P. aeruginosa* phage φE2005 has been observed in case of preventing the *P. aeruginosa* biofilm formation [206]. Another example of phage degradation of catheter-associated biofilm has been described recently [207]. In that study, the ability of phages to destroy biofilm formed by clinical isolates of multi-antibiotic-resistant *Providencia stuarti* on silicone and latex catheters was tested. The application of phages against *P. stuarti* led to a two-fold reduction in biofilm mass and a 2 log reduction in bacterial cells count [207].

Bacteriophages used for phage therapy in humans do not always produce the expected results, despite promising results obtained in vitro. A few commercially available phage cocktails registered in Georgia were tested in vitro against common urinary pathogens such as *E. coli* and *K. pneumoniae*. The lytic activity of these cocktails on the 41 tested *E. coli* strains varied from 66% (the cocktail named Pyo bacteriophage) to 93% (the cocktail named Enko bacteriophage). However, after adaptation of the Pyo bacteriophage cocktail, its lytic activity increased from 66% to 93% of tested strains, resembling the Enko bacteriophage cocktail in its efficiency [208]. The same commercial cocktail has been tested in a double-blinded clinical trial conducted in patients after prostatectomy. The effects of phage therapy in that study were not superior to those of the placebo group [209]. At the same time, they were not significantly worse than the effects obtained after standard antibiotic therapy. The phage cocktail used in that study consisted of multiple phages directed against the most common urinary pathogens. However, the titer of each phage was relatively low, between 10^4^ and 10^5^ pfu/mL [209].

Phage therapy can also be used in combination with antibiotic therapy. The efficacy of phages in reducing biofilm produced by a multi-antibiotic-resistant strain of *Acinetobacter baumannii* was tested in an in vitro urine model. The use of phages in combination with antibiotics resulted in a higher reduction of biofilm and persister cells than the use of antibiotics or phages alone [210]. Similar results were obtained in another study on biofilm-forming *Klebsiella* strains isolated from urine or urinary catheters [211]. Another example is an in vitro study, where the *E. coli* strain isolated from the urinary tract of infected patients was treated with a combination of phage and a low dose of ampicillin. Ampicillin used in sublethal concentrations (1/8 and 1/4 of MIC) together with a phage at an MOI of 10 resulted in a significantly higher growth reduction than antibiotic or phage used alone [212]. The synergy effect has been also observed in vivo. One of the pieces of evidence came from the case study of a 58-year-old patient after kidney transplant who developed a recurrent UTI with an extended-spectrum β-lactamase (ESBL)-positive *K. pneumoniae* strain. Even though this strain was carbapenem-susceptible, the antibiotic therapies failed seven times. Eventually, the persistent recurrent UTI has been successfully treated with a combination of antibiotic and phage preparation [213]. A similar case of a 63-year-old patient with a recurrent UTI caused by extensively drug-resistant *K. pneumoniae* showed that phage therapy alone resulted in relief of the patient’s symptoms [214]. However, *K. pneumoniae* cells were still detected in the urine at low levels. Nevertheless, treatment with phages in combination with otherwise non-active antibiotics led to the complete elimination of the pathogen from the urinary tract [214].

## 8. Bacteriophages in Female Reproductive System and Pregnancy

As indicated in Section 7, bacteriophages have been successfully used in treating UTI caused by *E. coli*, *P. aeruginosa*, or *K. pneumoniae* [160,208,215]. These bacteria are also known to be responsible for uterine infections, thus, phage therapy can also be used in such diseases [216,217]. Untreated or frequent uterine infections are linked with ectopic pregnancies, impaired ovarian functions, and infertility [217,218]. Furthermore, if infection occurs during pregnancy, it may result in preterm labor or other complications [216,219]. Pregnancy is a stage in female life where various physiological changes occur, and it results in a number of risks in relation to pharmacokinetics of different therapeutics [220,221]. The use of some medicines, including antibiotics, may also pose a risk to developing fetus [222,223]. Therefore, the use of phage therapy, which is generally considered safer than the use of antibiotics and lacking serious side effects, may be an attractive alternative for treating uterine infections as well as other diseases during pregnancy [216]. There have been trials of phage therapy effectiveness in the treatment of uterine infection in laboratory animals and cattle; however, there was no clear evidence of benefits [224,225]. On the other hand, the main concern of any type of therapy in pregnant females is substance transfer through the placenta and possible influence on the fetus.

The placenta is a temporary organ that facilitates oxygen, nutrient, and waste exchange between maternal and fetal circulations. It also serves as a protective barrier by reducing the entry of substances that can cause harm to the developing fetus. The transfer of these substances, including drugs, can be modified by metabolism and placental enzyme systems. However, nearly all drugs administered during pregnancy will eventually enter, to some degree, the fetal bloodstream via passive diffusion or by various active transporters [226]. Aside from chemical compounds, it has been shown that some microorganisms such as Listeria monocytogenes or Treponema pallidum and viruses such as HIV, Rubella, and Zika virus are also able to cross the placenta [227,228]. This leads to inquiry if phages are able to cross this barrier between woman and fetus, and if so, what are the consequences and how the phages are transferred.

First attempts to analyze phage penetration through the placenta were made in the 1920s. In vivo studies using guinea pig and rabbit models showed no placental crossing by coliphages and staphylococcal phages administrated orally via intravenous injection [229]. However, phages active against *E. coli* and *Bacillus* sp. were reportedly found in the bloodstream of newborn guinea pigs and placental samples after subcutaneous or intraperitoneal inoculation of mothers [229]. The passage of coliphages and mycobacteriophages across the placental barrier using a rat model was studied [230]. Methods of phage administration included intravenous injection and direct administration to the uterus. It was observed that injection into the uterine lumen resulted in phage presence in fetal body fluids. In case of intravenous administration, phages were detected in fetus only in individual cases [230]. Opposite results were also reported, where phage φX174 was shown to be present in fetal blood samples after intravenous inoculation of a mother [231]. However, it was found that the transfer depended on phage dosage, as φX174 was not detected in fetal blood if maternal phage concentration was below 10^7^/^mL^ [231]. Similar results were reported in another study, where T7 phage was successfully recovered from fetal samples 15 min after the mother was inoculated via tail vein injection [232].

The mechanism that phages utilize to cross the barrier between mother and fetus remains uncertain [216,230,231]. It was speculated that it may occur through splanchnopleure, as it is the primary transfer organ in some animals, or via syncytiotrophoblast layer [227,231]. It has been shown that this part of the placenta is particularly responsible for resistance against viral infections [227,228]. However, if not fully developed (early stages of pregnancy) or damaged, it can be crossed by viruses and potentially by phages [216,228,233,234]. Furthermore, it is possible that the level of phage penetration through the placenta as well as the passage mechanism may depend on the phage type and pregnancy stage [143,227,229,231]. Therefore, the safety of phage therapy in pregnant woman and phage influence on fetal safety is yet to be fully determined.

## 9. Interactions of Bacteriophages with Cancer

The first report on the effect of bacteriophages on cancer cells was presented by Bloch in 1940 (see ref. [235] for discussion). He observed the ability of phages to accumulate and inhibit the growth of Ehrlich malignancy. Studying the interactions between T4 and HAP1 phages and cancer cells, antimetastatic activity was demonstrated against a melanoma mouse model and Lewis lung cancer (LLC) model in vivo and inhibition of migration in mouse B16 and human Hs294T melanoma in vitro [19,236,237]. Another experiment showed enhanced antimetastatic efficacy when HAP1 and T4 phages were administered orally compared to intraperitoneal administration. This can be explained by the presence of *E. coli,* which is the host for both phages, in the intestinal microbiota. This allows bacteriophages to multiply in the gut [238]. Bacteriophage HAP1 was isolated from the T4 phage population. It exhibited a higher affinity for melanoma cells, and a nonsense mutation in the *hoc* gene has been identified which caused production of the truncated gene product by 44% of the original length. The *hoc* gene encodes the Hoc protein, which is a highly immunogenic outer capsid protein present on the T4 head surface. To investigate the impact of this mutation, the activity of T4 was compared to that of HAP1 and T2 phages (in which Hoc is not present). It was shown that the T2 and HAP1 phages degraded faster; nevertheless, their ability to inhibit metastasis was more effective than the T4 phage [19]. On the other hand, it was hypothesized that phage T4 antitumor activity is due to inhibition of the β3 integrin signaling pathway by the gp24 protein. This protein has a KGD (Lys-Gly-Asp) sequence that is homologous to the RGD-amino acid motif (Arg-Gly-Asp) capable of binding the β3 receptor. Integrins are expressed on the surface of many different cells, such as platelets. Their occurrence can also be observed in neoplastic cells. This is related to their increased ability to grow and form metastases. Examples of such integrins are αvβ3, αIIbβ3, and α5β1 [19,236,237,239]. In fact, phage binding to the surface of cancer cells was observed to be blocked by β3 receptor ligands and antibodies directed against β3 integrin [19].

Apart from interactions between T2, T4, or HAP1 bacteriophages and cancer cells, the activity of a lysate containing staphylococcal bacteriophages was tested. It was observed that there was a reduction in the migration of melanoma cells [240]. Another experiment was conducted on the HL-60 cell line using six phages, each specific for *Staphylococcus aureus, E. coli* (including T2, T4, HAP1), and *P. aeruginosa*. It was demonstrated that most phages had no effect on HL-60 migration except Staph.lysis (specific for *S. aureus*), which stimulated it [240]. Due to the lack of observations for the interaction of phages T4 and HAP1, which interacted with murine B16 melanoma cells and LLC, one might conclude that bacteriophages do not necessarily interact with every type of cancer.

The intrinsic signaling involved in the interaction between T4 and M13 phages and prostate cancer cells had also been investigated. The results showed the activation of some pathways such as PKB (AKT) and PI3K, as well as down-regulation of the Hsp90-encoding gene, which is involved in cell apoptosis [241]. By studying the interactions between the PK1A2 phage and neuroblastoma cells, a molecular mimicry was suggested. Due to the similarity of the receptor recognized on the host cell surface with the polysialic acid present on the surface of neuroblastoma cells, it is possible for phages to enter the eukaryotic cell. It has also been shown that phages are able to survive inside a eukaryotic cell for one day, with no effect on cell activity [12].

Another approach to study the interactions was to use modified M13 phages. Using in vivo panning, phage WDC-2 was isolated, which is specific for cancer cells in a mouse melanoma B16-F10. The second phage used was an HLA-A2 specific Fab-phage against established B16-F10 and B16/A2K^b^ tumors. Regression of tumors and prolongation of life in mice was observed [128]. This may suggest the activation of dendritic cells by bacteriophages, and resultant antigen presentation to tumor T lymphocytes. The recruitment of polymorphonuclear cells into tumor tissue might occur, and they could produce cytokines such as IFN-γ and IL-12. However, prior immunization with phages has not increased efficacy in cancer treatment [128]. Therefore, it can be suggested that phages may promote inflammation by activating TLR9 on antigen-presenting cells and neutrophils. When the MyD88^−/−^ splenocyte model, in which TLR signaling is abolished, was investigated, no tumor regression or neutrophilic infiltration was observed [128].

Tumor cells secrete various factors that enable recruiting tumor-associated macrophages (TAMs) with the M2 phenotype. Compared to M1-polarized macrophages, they secrete IL-10, TGF-β, and IDO factors, thus inhibiting the activation and proliferation of tumor-specific T cells. One approach of anti-cancer therapy is to polarize macrophages to the M1 phenotype. Such a mechanism was observed in C57BL/6 mice after tumor-specific phage therapy [127]. There was a 19-fold reduction in MMP-14 protease, which is responsible for the invasive activity of tumor cells. After the cessation of bacteriophage administration and complete tumor regression, it was noted that mice were protected from tumor recurrence in a CD8 T-cell-dependent manner [127]. Subsequent studies demonstrated inhibition of tumor growth by a CD8 T-cell-dependent mechanism. In some tumors, such as colorectal cancer, carcinoembryonic antigen (CEA) is overexpressed. Thus, an M13 phage capable of recognizing this antigen has been created [242]. When specific bacteriophages were administered to mouse models of colorectal cancer, increased expression of the costimulatory molecules CD80 and CD86 was observed in dendritic cells (DCs) present in lymph nodes and spleen [242]. These results indicated the activation of tumor-specific T cells by DCs.

Taking advantage of reports that phages can act as natural adjuvants, a BM-DCs (bone marrow-derived dendritic cells) vaccine was developed [243]. DCs were activated with T4 phage and then loaded with tumor antigens. The use of BM-DCs/T4 + TAg (tumor antigen) resulted in increased expression of differentiation markers and an enhanced ability to stimulate T cells to produce IFN-γ in C57BL/6 mice carrying advanced MC38 colon carcinoma tumors. Compared to controls, there was a delay in tumor growth of up to 19 days [243].

In studies conducted on the BL57C/6J animal model, in which mice carried transplanted Ehrlich carcinoma, the effectiveness of tumor treatment using phage therapy was compared with chemotherapy [244]. The mice were divided into five groups. The first group was the control group, the second was subjected to chemotherapy, the third was given permanent vaccination with phage lysate consisting of coliphages (*E. coli* Ph. L), and the fourth received chemotherapy and *E. coli* Ph. L. three times (at the 2nd, 6th, and 11th day of the experiment). The last group received chemotherapy and *E. coli* Ph. L. permanent vaccination. For the second group, it was noted that the tumor volume decreased relative to the control group, but the lifespan of mice was similar. The most effective tumor growth inhibition and lifespan extension was observed in animals in group five [244].

Cyclophosphamide (CTX) is a chemotherapeutic drug that can induce *Enterococcus hirae* translocation from the intestinal lumen to immune tissues of the mesentery and spleen. Comparing a panel of different *E. hirae* strains, it was found that only a few *E. hirae* isolates (13144 and IGR11) were effective in reducing MCA205 tumor size caused by CD8 T cell production in C57BL/6 mice. Then, the corresponding T cell epitopes were identified by comparing them to the sequences of different genes present in different *E. hirae* strains [245]. Using an in silico approach, 13144-specific nanopeptides with strong affinity for the MHC class I H-2k^b^ protein were identified. The in vitro study led to the identification of an epitope present in strain 13144 that was TSLARFANI (TMP1), which corresponds to a part of the amino acid sequence of the bacteriophage tail protein (TMP) [245]. Some tumors, such as MCA205 and TC1, overexpress the PSMB4 antigen, which is involved in tumor cell proliferation and invasion. Since the GSLARFRNI peptide belonging to the PSMB4 subunit shows strong homology with TMP1, molecular mimicry was observed, resulting in cross-reactivity between tumors and antigens in this phage [245].

## 10. Interactions between Prophages and Eukaryotic Cells

Lysogenic bacteria have bacteriophage genomes (called prophages) embedded in their chromosomes. In such cases, the mechanism called lysogenic conversion can occur, which is the acquisition of new characteristics by the bacterial cell. This allows prophages to indirectly affect eukaryotic cells. Such a conversion can be exemplified by toxins encoded by genes present in prophage genomes. One bacteriophage-encoded toxin is Shiga toxin, which is coded by genes present in lambdoid phages of lysogenic strains of Shiga toxin-producing *E. coli* (STEC) [246]. Expression of the toxin genes occurs as a result of prophage induction. One factor that can induce prophage is naturally occurring hydrogen peroxide. It is produced by neutrophils as a result of the immune response to the pathogen. Experiments were carried out to test whether its concentration is sufficient to induce prophages. The results showed that it can induce the excision of the prophage from the bacterial chromosome, leading at the same time to the release of the toxin through lysis of bacterial cells [246]. A more detailed description of the action of these phage-encoded toxins has been presented in Section 6.5.

The presence of prophages can also affect the quality of the biofilm produced by the bacteria. Biofilms allow bacteria to survive in adverse conditions and can hinder the access of antibiotics to cells. Biofilms formed by *Streptococcus pneumoniae* contain polysaccharides, proteins, and nucleic acids. It has been observed that the bacterial population gains from the lysis of single cells when a biofilm is formed [247]. Lysogenic strains were characterized by higher biomass and viability in biofilms compared to strains without prophage. These observations suggested that this phenomenon is due to the presence of external DNA (eDNA), which is formed by excision of the prophage and subsequent lysis of bacterial cells. It was also shown that the amount of eDNA in the biofilm of lysogenic bacteria is six times higher than that in the biofilm formed by bacteria without any prophage [247]. Another example is the T and B lymphocyte-stimulating protein (TspB), which is encoded by the *orf6* gene present in lysogenic *Neisseria meningitidis*. This protein is capable of binding human IgG, which explains the presence of antibodies in the biofilms formed [248]. The activating factor for IgG binding is most likely a protein concentrated in Cohn’s fraction IV [248]. Prophages may also affect bacterial gene transcription. The presence of the φMR11 prophage in the *Staphylococcus aureus* CC398 genome resulted in increased transcription of *clfA* and *fnbA* genes, coding for bacterial adhesins, which contribute to the colonization of heart valves [249].

Interactions between prophages and eukaryotic cells can also be seen by the presence of a eukaryotic association module (EAM) acquired by horizontal gene transfer in the genome of bacteriophage WO. The EAM has 10 domains, more than half of which show greater homology with eukaryotic than bacterial genes [21]. WO is a bacteriophage capable of integrating into the genome of *Wolbachia*, and this bacterium is capable of infecting invertebrates. The fusion of gametes of infected males and uninfected females with *Wolbachia* results in embryonic death caused by cytoplasmic incompatibility. This phenomenon leads to a delayed breakdown of the nuclear envelope in the pronucleus of the male, preventing normal chromosome condensation [250]. It was demonstrated that the prophage WO is responsible for the triggering of cytoplasmic incompatibility [251].

Lysogenic bacteria can acquire traits that directly affect the immune system of eukaryotes. Prophage SF370.1 encodes the extracellular DNase Spd1, causing DNA degradation in neutrophil extracellular traps (NETs), increasing *Streptococcus pyogenes* M1 invasiveness [252]. The gene encoding Spd1 is adjacent to the scarlet fever toxin gene, and their co-expression most likely occurs through prophage induction [252]. The presence of φNM1-4 prophages in the *S. aureus* genome in the Newman strain results in an increased acute immune response. Namely, non-lysogenic strains induced a cellular response with a 10-fold reduction in Th1 and Th17 cells relative to lysogenes [253].

Some bacteria possess a competence system (Com) that allows the uptake of naked DNA from the environment. Such a mechanism is called transformation. Com in *Listeria monocytogenes* 10403S is thought to be non-functional because a prophage φ10403S is incorporated in place of the functional *comK* gene. If lysogenic bacteria are located in the phagosome due to interaction with macrophages, excision of the prophage occurs, leaving the *comK* gene intact [254]. In the case of these bacteria, this gene is not able to carry out transformation but it can promote escape from phagosomes. No progeny viruses or bacterial cell lysis were observed during the experiments. It is suggested that this may be another unknown regulatory mechanism occurring in bacteriophages, or the bacteria are able to regulate phage survival by preventing cell lysis [254]. Another observation carried out in bacteria phagocytosed by macrophages showed induction and cell lysis of *E. coli*. The bacterial transcriptional regulator PhoP was shown to be a prophage inducer. It is a sensor of environmental stress including low Mg^2+^ levels, antimicrobial peptides, and acidic pH. It was suggested that phages may interact with the innate immune system (Figure 7) [255].

## 11. Concluding Remarks

Bacteriophages were thought for a long time to be totally neutral for eukaryotic cells, and especially for animals and humans, as they should be specific for bacterial cells. However, studies from recent years clearly indicated that phages can definitely interact with eukaryotes, either directly, by influencing cells, tissues, and organs, or indirectly, by modifying the content of microbiome. These bacterial viruses can significantly influence the functions of the immune system, respiratory system, central nervous system, gastrointestinal system, urinary tract, and reproductive system. They can also affect cancers. Their effects can be either beneficial or deleterious for animals and humans. Therefore, it is crucial to deeply understand mechanisms of interactions between bacteriophages and eukaryotes, which should allow us to wisely use these viruses in medicine (especially in phage therapy) and biotechnology, reasonably employing their properties and abilities to both combat bacteria and modulate functions of eukaryotic organisms.

## Figures and Tables

**Figure 1 ijms-22-08937-f001:**
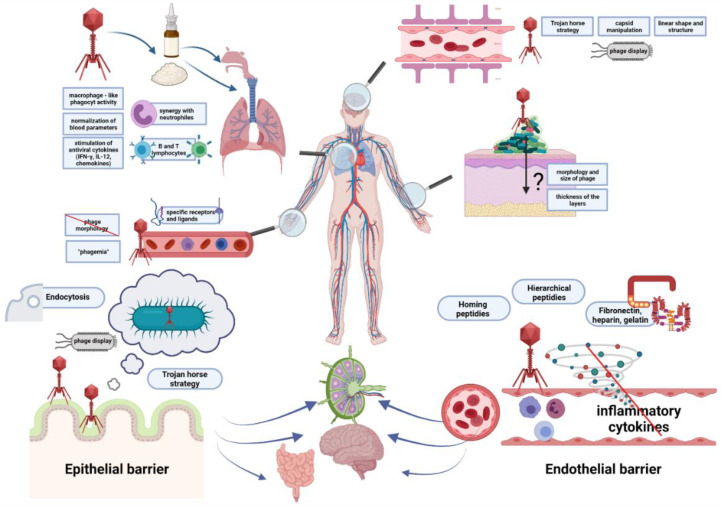
Barriers that must be crossed by bacteriophages to penetrate tissues and organs of animals and humans. Epithelial and endothelial barriers are shown in more detail in lower panels. Ways for penetration of bacteriophages to various tissues and organs are depicted and indicated by arrows. Specific strategies facilitating penetration, modifications of phages, and some effects of bacteriophages on animal and human organisms are presented in boxes. Bacteriophages, bacteria, and eukaryotic cells are shown as symbols mimicking their shapes, while proteins are marked as closed circles or other geometric figures.

**Figure 2 ijms-22-08937-f002:**
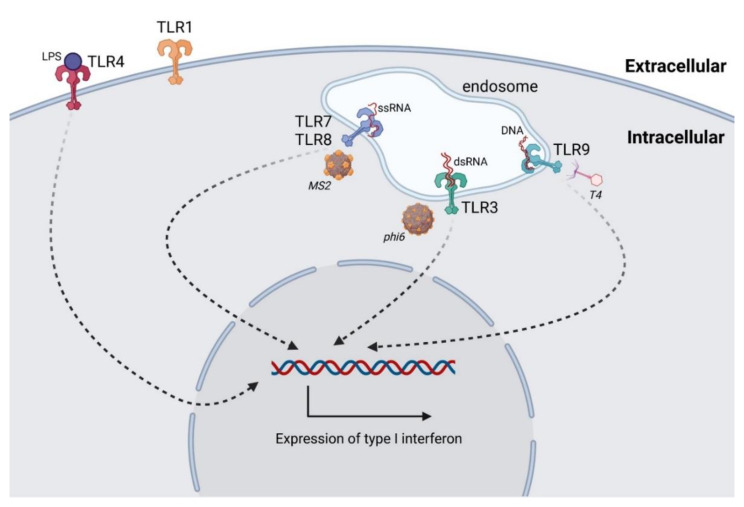
Intracellular interactions of bacteriophages with Toll-like receptors (TLRs). TLRs are capable of recognizing bacteriophage nucleic acids in endosomes (ssRNA (TLR7, TLR8), dsRNA (TLR3), and DNA (TLR9)). The scheme shows hypothetical interactions of bacteriophages with TLRs: bacteriophage MS2 (ssRNA), bacteriophage φ6 (dsRNA), and bacteriophage T4 (DNA). TLR4 can also be activated by LPS present in unpurified phage lysate. The activation of TLRs promotes the expression of type I IFNs.

**Figure 3 ijms-22-08937-f003:**
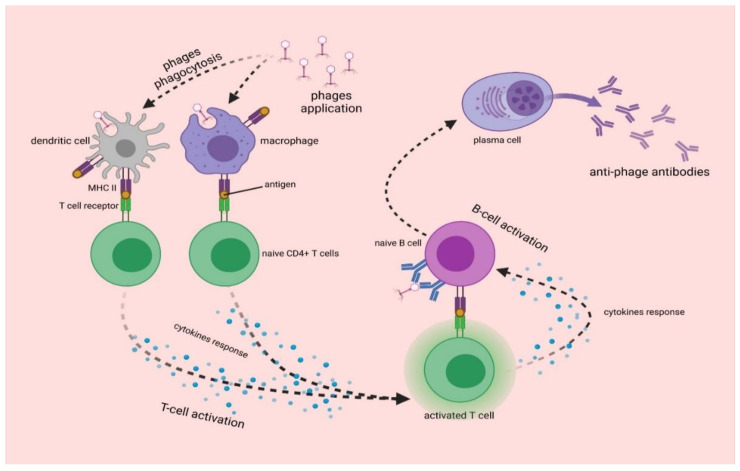
The synthesis of anti-phage antibodies. Upon entry into the body, bacteriophages are phagocytosed by dendritic cells and macrophages, which present phage antigen to naive CD4+ T cells. This leads to the activation of T cells, which present antigen to B cells. The differentiation of B cells results in the formation of plasma cells that produce anti-phage antibodies [8,105].

**Figure 4 ijms-22-08937-f004:**
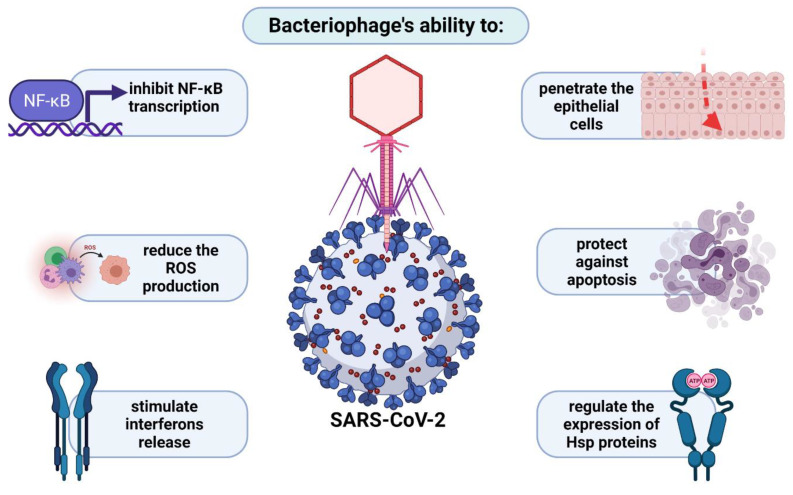
Bacteriophage potential in treatment of SARS-CoV-2 infections.

**Figure 5 ijms-22-08937-f005:**
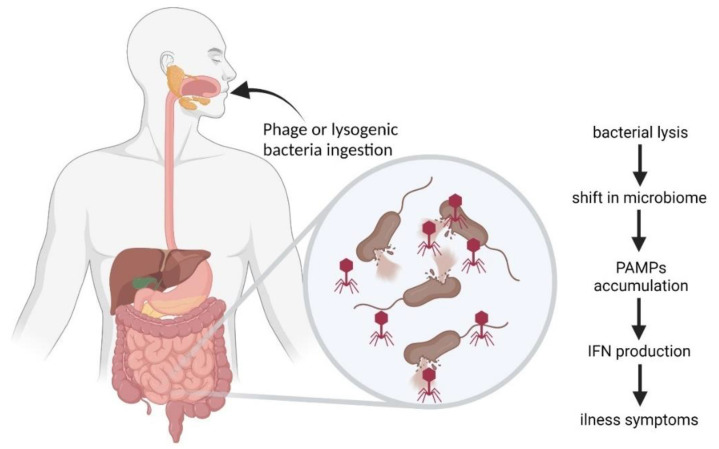
Scheme of “phage infection” as proposed by Tetz and Tetz [133] (modified).

**Figure 6 ijms-22-08937-f006:**
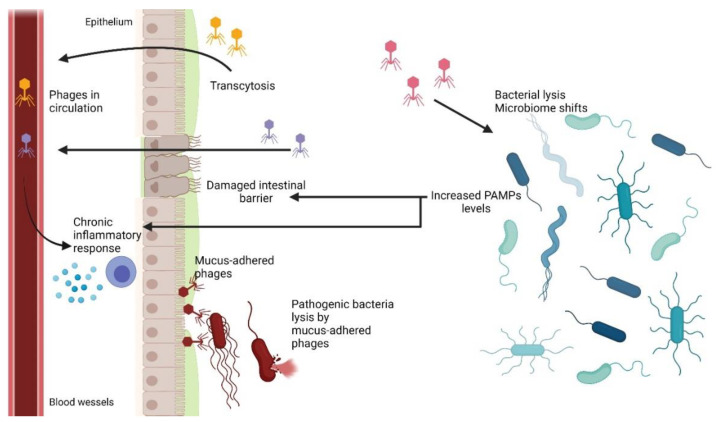
Concept of phage interactions with environment in intestine as proposed by Tetz and Tetz [133] and Barr et al. [138] (modified).

**Figure 7 ijms-22-08937-f007:**
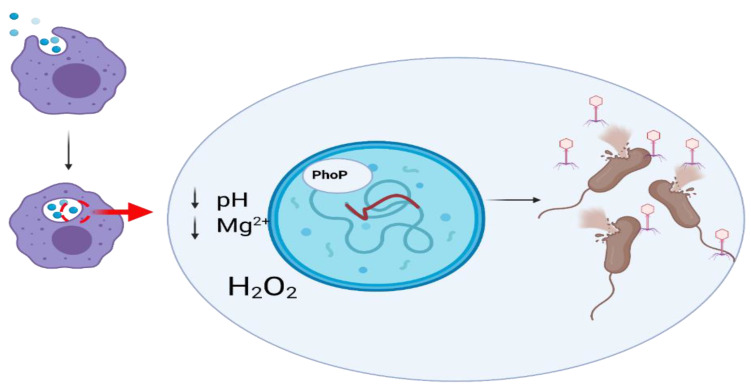
Interaction of lysogenic *E. coli* with macrophage. Recognition of pathogenic *E. coli* by macrophage results in phagocytosis. Inside the phagosome, specific conditions occur, such as low pH or low magnesium ion concentration. Hydrogen peroxide molecules are also present there. These conditions cause prophage induction through the action of the bacterial transcription regulator PhoP, leading to the lysis of *E. coli* cells and release of virions.

**Table 1 ijms-22-08937-t001:** Formation of antiphage antibodies after phage administration to mammals.

Bacteriophage	Detected Antibody	In Vivo Model	Source of Isolation	Administration Route	Reference
GACP	IgG, IgM	mice	blood	intraperitoneally	[111]
φ26, φ27, φ29	IgG, IgA	calf	sera	suppositories	[112]
bacteriophage specific to *E. coli*	undefined	rabbit	blood	subcutaneous injection	[113]
AbArmy ϕ1, AbNavy ϕ1, AbNavy ϕ2, AbNavy ϕ3, AbNavy ϕ4	IgG2a, IgG2b	mice	serum	intraperitoneally	[86]
A3R, 676Z	IgM, IgG	mice	plasma	per os	[114]

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
