# Peer review of "Interactions of Bacteriophages with Animal and Human Organisms—Safety Issues in the Light of Phage Therapy"

_ijms, 2021, doi:10.3390/ijms22168937_

Round 1

Reviewer 1 Report

The manuscript by M. Podlacha et al is well organized, presented in an intelligible fashion and written in common English. It covers different aspects of possible phage interactions with the higher eukaryotes and raises important new questions that need to be taken into account when developing phage-based therapies. I suppose this work would be interesting for a broad audience of professionals working in the field.

Reviewer 2 Report

The manuscript of Podlacha et al. describes the stay of the art about the interactions between phages and eukaryotic cells. The paper is sound and explains in detail the field, whose research has been scarce. I only have some minor comments:

- The introduction section I recommend just to be named as Introduction.

- Line 33 and others through the text should use double “ instead ,, at the beginning.

- Line 33 as make not sense

- Line 48. I would substitute ask for wonder

- Line 55. I would delete the word “intensively”

- Line 73. I would delete Many.

- Line 74. There is an error, it must be 2.1

- Line 81. First “ is missing

- Line 146. Please, specify homologs in what.

- Line 202. Please, explain a little how phage can lyse erythrocytes and leukocytes.

- The authors use past perfect instead past simple for describing old results, it must be changed, for example: line 275 (have described), line 284 (have demonstrated) and so on.

- Figure1 should include a short explanation of the figure.

- I would delete from Line 375 to line 380, it means the complete 3.1.1 Phagocytosis paragraph and I would add the text at the macrophages section. Moreover, I would delete the sentence “Detailed description of this process has been provided 379 above (Section 2.1).”

- Lines 424 and 425, please explain it better.

- Line 452. I would delete the sentence “Definitely, this 452 field requires further studies.”

Author Response

REVIEWER'S COMMENT:

The manuscript of Podlacha et al. describes the stay of the art about the interactions between phages and eukaryotic cells. The paper is sound and explains in detail the field, whose research has been scarce. I only have some minor comments:

REVIEWER'S COMMENT:

The introduction section I recommend just to be named as Introduction.

RESPONSE:

CORRECTED AS SUGGESTED BY THE REVIEWER

REVIEWER'S COMMENT:

- Line 33 and others through the text should use double “ instead ,, at the beginning.

RESPONSE:

CORRECTED AS SUGGESTED BY THE REVIEWER

REVIEWER'S COMMENT:

- Line 33 as make not sense

RESPONSE:

“as” WAS SUBSTITUTED FOR “because”

REVIEWER'S COMMENT:

- Line 48. I would substitute ask for wonder

RESPONSE:

“ask” WAS SUBSTITUTED FOR “wonder”

REVIEWER'S COMMENT:

- Line 55. I would delete the word “intensively”

RESPONSE:

CORRECTED AS SUGGESTED BY THE REVIEWER

REVIEWER'S COMMENT:

- Line 73. I would delete Many.

RESPONSE:

CORRECTED AS SUGGESTED BY THE REVIEWER

REVIEWER'S COMMENT:

- Line 74. There is an error, it must be 2.1

RESPONSE:

CORRECTED AS SUGGESTED BY THE REVIEWER

REVIEWER'S COMMENT:

- Line 81. First “ is missing

RESPONSE:

CORRECTED AS SUGGESTED BY THE REVIEWER

REVIEWER'S COMMENT:

- Line 146. Please, specify homologs in what.

RESPONSE:

It is indicated now that there are homologs of fragments of various genes (line 146).

REVIEWER'S COMMENT:

- Line 202. Please, explain a little how phage can lyse erythrocytes and leukocytes.

RESPONSE:

THIS WAS AN ERROR – WE THANK REVIEWER FOR THIS NOTE. THE CORRECTED SENTENCE READS AS FOLLOWS: An important point is the ability of coliphages to adhere to erythrocytes as well as leukocytes (line 202)

REVIEWER'S COMMENT:

- The authors use past perfect instead past simple for describing old results, it must be changed, for example: line 275 (have described), line 284 (have demonstrated) and so on.

RESPONSE:

CORRECTED AS SUGGESTED BY THE REVIEWER

REVIEWER'S COMMENT:

- Figure1 should include a short explanation of the figure.

RESPONSE:

AS REQUESTED BY THE REVIEWER, THE LEGEND TO FIGURE 1 HAS BEEN EXTENDED. THE MODIFIED VERSION READS AS FOLLOWS: Figure 1. Barriers that must be crossed by bacteriophages to penetrate tissues and organs of animals and humans. Epithelial and endothelial barriers are shown in more detail in lower panels. Ways for penetration of bacteriophages to various tissues and organs are depicted and indicated by arrows. Specific strategies facilitating penetration, modifications of phages, and some effects of bacteriophages on animal and human organisms are presented in boxes.

REVIEWER'S COMMENT:

- I would delete from Line 375 to line 380, it means the complete 3.1.1 Phagocytosis paragraph and I would add the text at the macrophages section. Moreover, I would delete the sentence “Detailed description of this process has been provided 379 above (Section 2.1).”

RESPONSE:

CORRECTED AS SUGGESTED BY THE REVIEWER

REVIEWER'S COMMENT:

- Lines 424 and 425, please explain it better.

RESPONSE:

AS SUGGESTED BY THE REVIEWER, THIS TEXT HAS BEEN MODIFEID. THE REVISED FRAGMENT READS AS FOLLOWS: However, number of phage particles per ml was similar in experiments with phages administered alone and in combination with LPS. This indicated that under these conditions, there were no effects of LPS on macrophage phagocytic potential. The most probable explanation for this phenomenon was the activation of macrophages by bacteriophages through a signaling pathway different than LPS [64]. (lines 422-427).

REVIEWER'S COMMENT:

- Line 452. I would delete the sentence “Definitely, this 452 field requires further studies.”

RESPONSE:

CORRECTED AS SUGGESTED BY THE REVIEWER

Reviewer 3 Report

Dear authors, 

In my opinion, the authors summarizing the works about interactions between bacteriophages and eukaryotes created a currently sorely lacking publication. I recommend the work for publication after introducing minimal changes.

First of all, please make sure that in the final version of the manuscript, bacterial names are in italic - through the text, I found many non-italicized bacterial names.

Please make sure that statements "A few commercially available 1121 phage cocktails registered in Georgia were tested in vitro against common urinary path- 1122 ogens such as E. coli and K. pneumoniae. The lytic activity of these cocktails on 41 of tested 1123 E. coli strains varied from 66% to 93%. However, after adaptation of one of these cocktails 1124 (named Pyo), its lytic activity increased from 66% to 93% of tested strains [212]" is correct, the numbers 66% and 93% are repeated.

Author Response

REVIEWER'S COMMENT:

In my opinion, the authors summarizing the works about interactions between bacteriophages and eukaryotes created a currently sorely lacking publication. I recommend the work for publication after introducing minimal changes.

First of all, please make sure that in the final version of the manuscript, bacterial names are in italic - through the text, I found many non-italicized bacterial names.

RESPONSE:

WE HAVE REVISED THE TEXT, AND ALL NAMES OF BACTERIAL SPECIES AND GENERA ARE IN ITALIC.

REVIEWER'S COMMENT:

Please make sure that statements "A few commercially available 1121 phage cocktails registered in Georgia were tested in vitro against common urinary path- 1122 ogens such as E. coli and K. pneumoniae. The lytic activity of these cocktails on 41 of tested 1123 E. coli strains varied from 66% to 93%. However, after adaptation of one of these cocktails 1124 (named Pyo), its lytic activity increased from 66% to 93% of tested strains [212]" is correct, the numbers 66% and 93% are repeated.

RESPONSE:

WE HAVE MODIFIED THIS TEXT, AND THE REVISED VERSION READS AS FOLLOWS: The lytic activity of these cocktails on 41 of tested E. coli strains varied from 66% (the cocktail named Pyo bacteriophage) to 93% (the cocktail named Enko bacteriophage). However, after adaptation of the Pyo bacteriophage cocktail, its lytic activity increased from 66% to 93% of tested strains, resembling the Enko bacteriophage cocktail in its efficiency [212].